# Multi-omics in nasal epithelium reveals three axes of dysregulation for asthma risk in the African Diaspora populations

Asthma has striking disparities across ancestral groups, but the molecular underpinning of these differences is poorly understood and minimally studied. A goal of the Consortium on Asthma among African-ancestry Populations in the Americas (CAAPA) is to understand multi-omic signatures of asthma focusing on populations of African ancestry. RNASeq and DNA methylation data are generated from nasal epithelium including cases (current asthma, N = 253) and controls (never-asthma, N = 283) from 7 different geographic sites to identify differentially expressed genes (DEGs) and gene networks. We identify 389 DEGs; the top DEG, *FN1*, was downregulated in cases (q = 3.26 × 10$^{-9}$) and encodes fibronectin which plays a role in wound healing. The top three gene expression modules implicate networks related to immune response (*CEACAM5*; p = 9.62 × 10$^{-16}$ and *CPA3*; p = 2.39 × 10$^{-14}$) and wound healing (*FN1*; p = 7.63 × 10$^{-9}$). Multi-omic analysis identifies *FKBP5*, a co-chaperone of glucocorticoid receptor signaling known to be involved in drug response in asthma, where the association between nasal epithelium gene expression is likely regulated by methylation and is associated with increased use of inhaled corticosteroids. This work reveals molecular dysregulation on three axes – increased Th2 inflammation, decreased capacity for wound healing, and impaired drug response – that may play a critical role in asthma within the African Diaspora.

Asthma is the most common disease of childhood and its prevalence continues to increase in many parts of the world[1]. Despite advances in therapeutics and a better understanding of environmental risk factors, disparities between populations are profound, and cannot be entirely explained by non-genetic factors[2–5]. In the U.S., childhood asthma prevalence is 20% higher among African Americans compared to non-Hispanic children of European descent, and African American children are three times more likely to die from asthma[6]. Individuals of African ancestry have greater asthma morbidity and mortality both within[2] and outside the U.S.[7–9], and asthma prevalence is high in African countries and countries with populations of African ancestry[9–11]. There are notable differences in therapeutic response[12]; patients of African ancestry respond differently to inhaled corticosteroids (ICS)

compared to patients of European ancestry[13,14], and long acting beta agonizts (LABA) may have greater risk for adverse events in this group[15]. Genetic variants in the receptors for pharmaceutical agents used in asthma management may partially explain these differences in response[12]. To address gaps related to the genomic underpinnings of disparities in asthma, we established the Consortium on Asthma among African ancestry Populations in the Americas (CAAPA)[16].

Genome-wide association studies (GWAS) have identified >170 loci for asthma[17]. CAAPA has contributed the single largest GWAS focused solely on the African Diaspora[18], but populations of African ancestry remain woefully under-represented in large scale international studies[19,20]. In the recent Global Biobank Meta-analysis Initiative (GBMI), African ancestry individuals represented a mere 5% of the total

✉ e-mail: kathleen.barnes@cuanschutz.edu; rmathias@jhmi.edu

number of asthma cases[19]. Nonetheless, the ancestry diversity in the GBMI facilitated identification of SNPs with different effects across ancestries and increased power to identify loci missed in European-only subsets. Importantly, the GBMI demonstrated that increased diversity and sample size of the discovery GWASs were primary drivers of improved polygenic risk score (PRS) accuracy in non-European populations.

While there is mounting evidence that asthma risk variants may play a role in the regulation of immune system pathways[20,21], molecular mechanisms underlying asthma heterogeneity and their signals at GWAS loci are poorly described. With a few exceptions like the Chr17q locus, GWAS loci have offered limited insight into the biology of asthma, and even less into the complex heterogeneity and disparities in asthma. Transcriptomic approaches have demonstrated some success in profiling asthma endotypes[22] and epigenetics can provide mechanistic understanding of the regulation of these transcriptomic signatures[23], especially in the context of environment[24,25]. There are now multiple lines of evidence supporting the important role of the airway epithelium in the pathogenesis of asthma[26]. A meta-analysis of gene expression from airway epithelial cells has identified >400 differentially expressed genes, consistency between signatures from upper (nasal) and lower (bronchial) airway epithelial cells, and upregulation of Th2 pathway genes[26] – pathways related to allergen specific T-helper 2 (Th2) cells that are centrally involved in allergic asthma. Transcriptomic signatures can distinguish "type 2 high" from "type 2 low" endotypes of asthma, and importantly, such signatures are now shown to be associated with response to ICS[27], with the potential to also distinguish viral and non-viral asthma exacerbations[28]. Monoclonal antibodies including anti-IgE, anti-IL5, anti-IL5 receptor, and anti-IL4 receptor have advanced the management of severe asthma, with the choice of therapy predicated on the ability to define the endotype of asthma in the patient[29].

We hypothesize that transcriptomic signatures from the nasal airway epithelium in asthma cases and controls representing the African Diaspora will allow us to validate previously identified gene expression signatures of asthma and, importantly, identify pathways of dysregulation that are relevant to the disparities observed with respect to asthma. We rely on nasal epithelium as a proxy for the airways given its ease of tissue collection on large numbers of individuals and the established correlation between signatures of asthma between nasal epithelium and bronchial tissue[26,30]. RNA sequencing and DNA methylation data from the same nasal epithelium samples in cases (individuals with current asthma) and controls (individuals never having asthma) representing 7 locations across the African Diaspora reveals dysregulation on three axes – increased Th2 inflammation, decreased capacity for wound healing in airway epithelium, and impaired drug response – that play a role in the development of asthma in individuals of African ancestry.

## Results
### Clinical characteristics
Cases ($N = 253$) with current asthma status and controls with never-asthma status ($N = 283$) were recruited from seven sites including 4 US-based locations and 3 international locations (Supplementary Data 1). The subjects were 70% adult and 38% male. Global genetic ancestry deconvolution revealed a wide range in African ancestry (YRI ancestry = 9–100%) representing diversity across the African Diaspora (Supplementary Data 1, Supplementary Fig 1). The highest African ancestry was observed in the subjects from Nigeria (average = 100% YRI), and the lowest was observed in the subjects from Brazil (average = 51% YRI). Cases had higher total serum IgE ($p = 1.66 \times 10^{-23}$), higher eosinophil counts ($p = 3.42 \times 10^{-05}$) and higher phadiotop ($p = 2.27 \times 10^{-21}$), reflecting the greater burden of the allergic phenotype in this group. Cases also had poorer lung function (FEV1; $p = 7.30 \times 10^{-16}$). Similar patterns were observed within each site

comparing cases to controls, with differences also observed between sites. Asthma cases from all the three non-US recruitment locations had higher IgE compared to Chicago, but no difference in IgE levels was noted for the controls (Supplementary Data 1). Notably, cases from Salvador, Brazil had the most severe asthma (CASI score average = 8), and the accompanying highest rate of medication use (97.2%).

### Single gene analysis
We identified 389 differentially expression genes (DEGs) between cases and controls (Supplementary Data 2); 41% of these DEGs had at least 2 sites with site-specific $p < 0.05$ (Fig. 1A). We evaluated replication of these signals in a published meta-analysis of 8 studies including nasal epithelium transcriptomics[26]. A total 16,269 genes were tested in both analyses, including 353 of the 389 DEGs we identified; 87 of the 353 genes were independently identified in the meta-analysis ($q < 0.05$, enrichment $p$-value = $9.97 \times 10^{-27}$) and all but one showed the same direction of effect (Supplementary Data 2). An additional 64 genes were replicated at a nominal threshold ($p < 0.05$) in the meta-analysis with the same direction of effect (Supplementary Data 2). The 15 most significant DEGs (Supplementary Data 3, Supplementary Fig. 2) included genes known to play a role in wound healing (*FN1, CDH11*), immune response (*VSIG4, HS3ST4*), and asthma drug response (*PTHHD4, SPTBN1, FKBP5*). Additionally, *SNTG2* is the target of multiple miRNAs related to asthma[31], and *PPP1R9A* expression was previously determined to be influenced by IL-13 in mouse lung[32]. Effect sizes for the top 15 genes were generally consistent across the seven recruitment sites and, with the exception of *RHEX2*, and effect sizes were similar with overlapping confidence intervals between adult and pediatric subset analyses (Supplementary Fig. 2).

The most significant DEG was *FN1* (log2FC = −0.62, $q = 3.26 \times 10^{-9}$, Supplementary Data 3). Despite variability in *FN1* expression across sites (Fig. 1B) there was consistently lower expression in cases (Fig. 1B), similar effect sizes in the adult and pediatric subsets, and consistent effects across all sites except Washington DC (Fig. 1C, D). *FN1* was independently identified by Tsai et al.[26] with a similar lower expression in asthma ($z = −3.61$, $q = 1.25 \times 10^{-2}$).

### Pathway analysis on significant DEGs
Ingenuity Pathway Anlaysis (IPA) analysis on the $N = 389$ significant DEGs identified 1188 upstream regulators with p-value of overlap <0.05; the set of top 10 upstream regulators with notable roles in asthma are indicated in Supplementary Fig 3 and Supplementary Data 4. These include inflammatory cytokines *IL4* ($z = 0.096$, $p = 7.25 \times 10^{-10}$, Supplementary Fig 3B) and *TGFβ1* ($z = −1.711$, $p = 5.47 \times 10^{-8}$, Supplementary Fig 3C), both of which are known to play key roles in asthma. Interestingly, two asthma drugs – dexamethasone ($z = 2.117$, $p = 4.31 \times 10^{-10}$, Supplementary Fig 3D) and fluticasone propionate ($z = 1.44$, $p = 9.42 \times 10^{-08}$, Supplementary Fig 3E) – were also among the top 10 upstream regulators for the significant DEGs. The 188 genes identified as direct or indirect targets of these two drugs were not themselves related to medication use in CAAPA (Supplementary Data 5). Beyond its role in immune-related mechanisms for asthma, *TGFβ1* has also been found to be a potent stimulus for *FN1* expression in vascular and airway smooth muscle (ASM) cells, lung fibroblasts, and the alveolar epithelial cell lines[33]. In our study, it was identified as an upstream regulator for a network that includes three genes that are known to play a role together in airway remodeling and wound healing – *FN1, COL3A1* and *COL41* (Supplementary Fig 3C)[34]. All three genes have lower nasal epithelial expression among asthmatics (*FN1* (log2FC = −0.62, $q = 3.26 \times 10^{-9}$); *COL3A1* (log2FC = −0.32, $q = 3.54 \times 10^{-2}$); *COL4A1* (log2FC = −0.24, $q = 3.90 \times 10^{-2}$), Supplementary Data 2).

### Gene expression module analysis
There were 24 weighted gene correlated network analysis (WGCNA) modules identified from analysis of $N = 1326$ genes (DEGs with FDR <

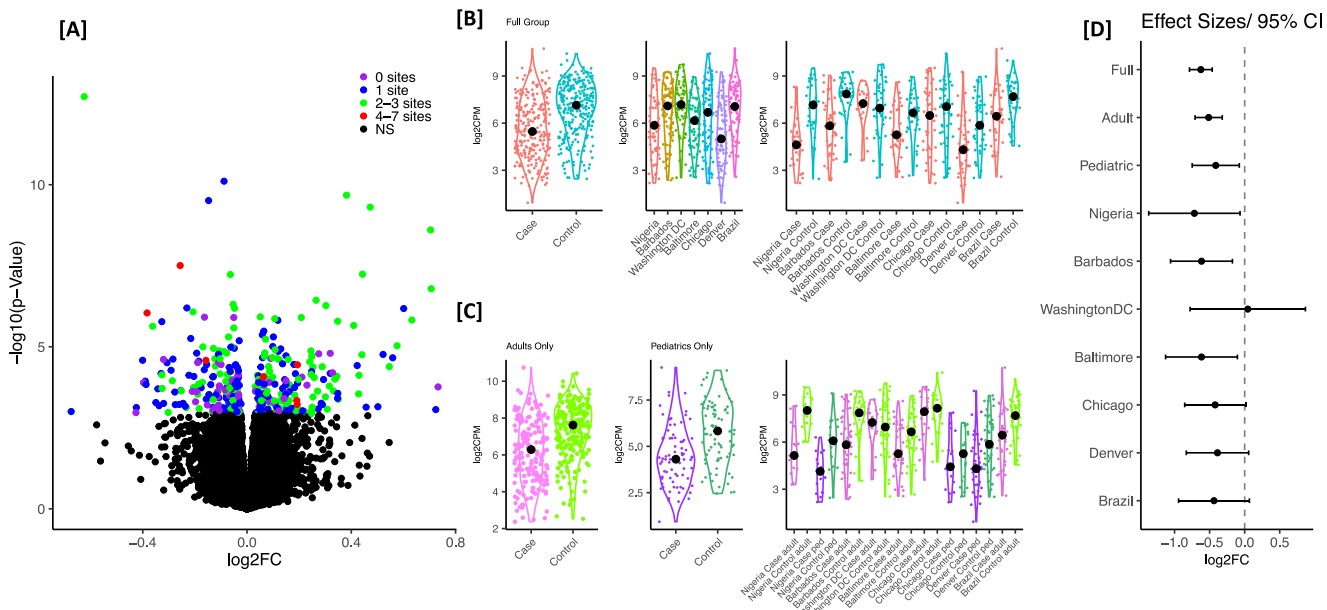

**Fig. 1 | Summary of the DEG analysis for active asthma in CAAPA. Panel A** Volcano plot of DEG analysis for asthma in the full combined group ($N = 253$ cases, $N = 283$ controls) from all 7 sites. Color represents the number of sites where the uncorrected significance for the DEG analysis within the site was $p$-value < 0.05, and genes that did not cross FDR of 0.05 in full combined analysis are retained as black. **Panel B** Combined gene expression for top gene *FN1* by site. **Panel C** Gene

expression for top gene *FN1* stratified by adult vs. pediatrics. **Panel D** DEG effect sizes (log2 fold change and the 95% confidence interval) for top gene FN1 looking at the full combined analysis, analysis stratified by adults vs. pediatrics and the analysis within each site. CAAPA sites are ordered based on average African ancestry (% YRI) from highest (Nigeria) to lowest (Brazil). The test used in the DEG analysis was a moderated two-sided t-statistic. Source data are provided as a Source Data file.

0.15). Of these, 16 modules ranging in size from $N = 21$–88 genes were significantly differentially expressed by asthma status (Table 1, Fig. 2); gene-module membership is shown in Supplementary Data 2. We note that none of these $N = 1326$ genes were differentially expressed based on nasal steroid (NS) usage 5 days prior to nasal epithelium sampling in cases (minimum $q$-value = 0.89 comparing 32 cases on NSs but not withholding compared to 36 cases on NS that withheld NS; Supplementary Data 6). Of the five modules upregulated in asthma cases, the strongest was a network (M6) with *CEACAM5* as the hub gene (Fig. 2, Supplementary Fig 5A, log2FC = 0.32, $q = 9.62 \times 10^{-16}$). Also strongly upregulated in asthma cases, was the network (M2) with hub gene *CPA3* (Fig. 2, Supplementary Fig 5B, log2FC = 0.54, $q = 2.39 \times 10^{-14}$). In pathways downregulated in asthma cases, hub genes were identified reflecting the importance of impaired wound healing: *FN1* (Fig. 2B) as noted above, and *ERBB2* [35,36]. Additionally, networks related to drug response are implicated with hub genes *DNAH5*[37](M9), *NCALD*[38](M4), and *ST13*[39](M11) from downregulated modules for asthma.

The WGCNA modules reflected the same three axes of dysregulation in asthma cases as indicated from the top 15 DEGs and IPA upstream regulators on 389 DEGs; it should be noted that many of the associated modules and therefore the axes of dysregulation are correlated to varying degrees (Fig. 2A). The cumulative effect of these three axes is illustrated in a joint model examining dichotomized module expression focusing on the most significant module for each axis: Th2 inflammation (*CEACAM5*,M6), wound repair (*FN1*,M5), and drug response (*NCALD*,M4) (Fig. 2C). These modules are significantly correlated (M6-M4: R = −0.70, $p < 0.001$/M5-M6: R = −0.58, $p < 0.001$, and M5-M4: R = 0.29, $p < 0.001$). The ORs for asthma if an individual was in the upper median for any one, two and all three modules were 1.70 (95% CI 1.49–1.99), 2.91 (2.12–3.98), and 4.95 (3.09–7.93), respectively. This joint analysis provides evidence the greatest risk for asthma was when there is dysregulation along all three axes (Fig. 2D). The number of modules significant in the additive logistic model ($p = 2.8 \times 10^{-11}$) and no departures from additivity were detected ($p = 0.29$ from a 2df likelihood ratio test).

## Integration of gene expression with DNA methylation (DNAm)

There were 8,418 eQTM tests performed for gene-CpG pairs comprising significant DEGs and CpGs mapping within 5 kb of the gene transcription start site or that were annotated by promoter-capture HiC in bronchial epithelial cells lying in putative enhancer regions for these genes. Of these, 918 gene-CpG pairs had uncorrected eQTM $p < 0.05$ (Supplementary Data 7); this included 288 unique genes and 915 unique CpGs. Testing for differential DNAm by asthma status, we found only five of these CpGs to be DMCs (i.e. significant for asthma at the Bonferroni level of $p < 0.05/915$, Supplementary Data 8): two CpGs for *FKBP5* (cg03546163, cg23416081), two for *TREML2* (cg26928682, cg18297196) and one for *TMEM71* (cg27159719). We found cg03546163 to be the strongest predictor of *FKBP5* expression and cg26928682 for *TREML2* expression independent of other eQTMs for each gene. For *FKBP5* (Fig. 3A,B), adjusting for methylation at cg03546163 strongly attenuated the association between gene expression and asthma, with a reduction in effect size (log2FC = 0.231 and 0.106 pre- and post-adjustment for methylation at cg03546163), and a loss in significance ($p = 0.0019$ and 0.155 pre- and post-adjustment for methylation at cg03546163). The same pattern was noted for *TREML2* (Supplementary Fig 6) where the DEG lost significance after adjusting for methylation at cg26928682. However, the *TMEM71* asthma DEG remained largely unchanged when adjusting for methylation at cg27159719 (Supplementary Fig 6). These results suggest an epigenetic mechanism of regulation of gene expression in asthma risk for both *FKBP5* and *TREML2*.

The two CpGs recognized as eQTMs for *FKBP5* map to regulatory features, and there is interaction between the two regulatory features possibly accounting for the correlation observed between the two CpGs and asthma risk (Fig. 3C). *FKBP5* encodes FKBP prolyl isomerase 5, a cochaperone modulating glucocorticoid receptor (GR) activity associated with the inflammatory response[40]. While site-specific effects of gene expression at *FKBP5* with asthma are consistent with overlapping confidence intervals, a markedly stronger effect was observed in the Brazil site (Supplementary Fig 2) which is also the site

**Table 1 | Significant WGCNA gene module differential expression analysis for asthma**

| Module label | Hub gene | | log2 FC | Adjusted p-value | # genes in module | # genes with DEG q < 0.05 | Most signifiicant |
|---|---|---|---|---|---|---|---|
| | WGCNA | STRING | | | | | DEG in module |
| M6 | **CEACAM5** | MET | 0.32 | 9.62E−16 | 71 | 26 | HS3ST4 |
| M2 | **CPA3** | POSTN | 0.54 | 2.39E−14 | 81 | 45 | PTCHD4 |
| M5 | DKK3 | **FN1** | −0.28 | 7.63E−09 | 72 | 34 | FN1 |
| M20 | C16orf89 | RIMS1,PPP1R9A,NR2F1 | −0.36 | 3.41E−08 | 33 | 14 | PPP1R9A |
| M23 | **ACVR1B** | CCND1 | −0.16 | 4.56E−08 | 23 | 5 | SUSD4 |
| M15 | **SF3A1** | SMARCA2 | −0.06 | 1.61E−05 | 44 | 18 | SPTBN1 |
| M1 | PDCD10 | EEF1E1 | 0.07 | 3.74E−05 | 88 | 21 | ETAA1 |
| M4 | IFT172 | GLRB,**NCALD**,TP53BP1,PTPRT | −0.13 | 3.74E−05 | 77 | 17 | SLC13A3 |
| M3 | ENSG00000279476 | BCAN,PLAGL1,**UGT1A1**,PRRT2,GARNL3,CYP2A7 | 0.16 | 1.24E−04 | 78 | 15 | ENSG00000273599 |
| M21 | POLD2 | POLD2,UBB | −0.09 | 2.17E−04 | 31 | 11 | GNAS |
| M9 | DNAH10 | **DNAH5** | −0.13 | 2.46E−04 | 58 | 15 | DNAH5 |
| M24 | RPL7A | RPL23A | −0.08 | 2.29E−03 | 21 | 5 | MT3 |
| M14 | SARS1 | VCP | −0.05 | 2.58E−03 | 44 | 11 | POFUT1 |
| M11 | **ST13** | HSP90AB1 | −0.04 | 4.15E−03 | 48 | 9 | FAM169A |
| M8 | PRKCSH | **ERBB2** | −0.05 | 6.25E−03 | 62 | 18 | VPS18 |
| M10 | ASPM | BIRC5,CCNA2 | 0.10 | 2.74E−02 | 48 | 11 | PARPBP |

Genes with q < 0.15 in full group DEG analysis for asthma (see Supplementary Table S1) were grouped into 24 modules using WGCNA. 24 modules were tested for association with asthma, and the 16 identified with q < 0.05 are shown in this table. Bold font indicates gene used to label the module in Fig. 2. Analysis was performed using a moderated t-statistic (two-sided).

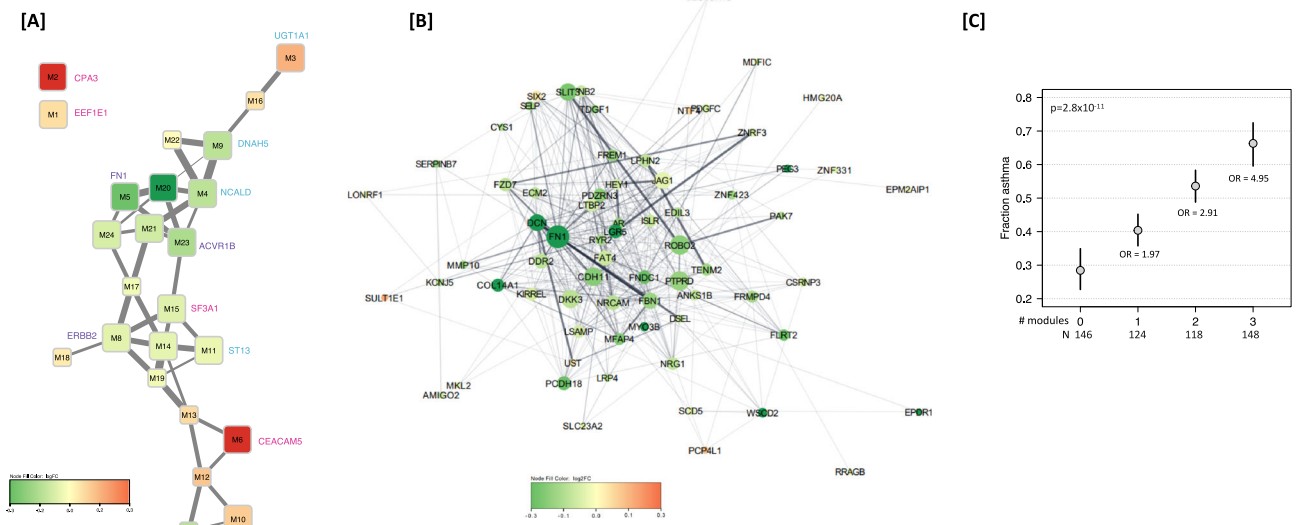

**Fig. 2 | Differential module expression based on the N = 1326 DEGs with FDR < 0.15 for active asthma. Panel A** Module connectivity network for the 24 modules. Each node represents a module, and each edge represents a significant positive Pearson pairwise correlation of module expression (correlation >0.5). Node color intensity corresponds to log2FC in DE Module analysis for asthma (red upregulated in cases, green downregulated in cases). Differentially expressed modules are larger in size (q < 0.05). Edge weight indicates correlation (wider edges higher correlation of module expression). **Panel B** STRING network retrieved for genes assigned to module M5 with hub gene *FN1*. Each node represents a gene and each edge represents a protein-protein interaction with a stringdb score >0.15. Node color intensity corresponds to log2FC in DE analysis of asthma (red upregulated in cases, green downregulated in cases). Node size was made proportional to the number of interactions of the node divided by maximum number of interactions of a node in the gene module (dg/max dg of module). Unconnected nodes were not included. Edge weight and transparency indicate stringdb score (wider, darker edges indicate higher score). **Panel C** Fraction of asthma cases and ORs for asthma if an individual was in the upper median for any one, any two and all three modules (M4, M5, M6). Fitted probabilities (gray dots) and 95% confidence intervals (black lines) were derived from a logistic model with number of modules as an additive predictor. Source data are provided as a Source Data file.

with the most severe asthmatics and highest inhaled corticosteroid (ICS) use in asthma cases (97.2%, Supplementary Data 1). *FKBP5* is not differentially expressed by ICS use in the full CAAPA dataset (Supplementary Data 5, p = 0.615). However, given prior evidence that *FKBP5* may be differentially expressed after dexamethasone treatment[41], we performed analyses to tease apart the relationship between ICS use, methylation at cg03546163, asthma severity by CASI, and risk for

asthma in the four US sites where both DNAm and RNAseq were available. Lower methylation at cg03546163 was strongly associated with asthma (β = −0.295, p = 8.2 × 10$^{-7}$) in the full subset of asthma cases and controls (N = 331). When limited to those asthma cases not on moderate-high ICS use (N = 283), the association with asthma remained (β = −0.197, p = 1.04 × 10$^{-3}$). In a subset case-only analysis of all asthmatics (N = 149), we observed a more significant association

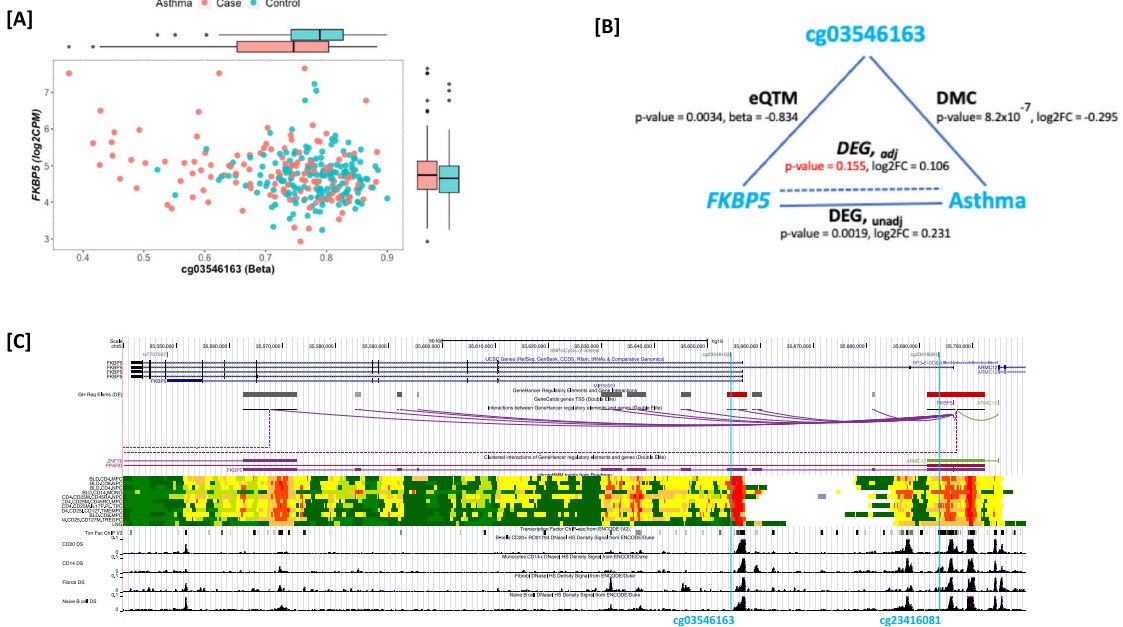

**Fig. 3 | Epigenetic mechanism relating gene expression to asthma for *FKBP5*.**
**Panel A** Scatter plot of methylation (beta) values at cg03546163 vs gene expression (log2 CPM) values for *FKBP5* and box plots showing median, lower and upper quartiles, whiskers extending to the furthest data point no more than 1.5 times the distance between the lower and upper quartiles, and outliers, by asthma case and control status for *N* = 298 individuals. **Panel B** Effect sizes and unadjusted p-values from two-sided multivariate linear regression models for DMC analysis (cg03546163 and asthma, *N* = 331), eQTM analysis (cg03546163 and *FKBP5* expression, *N* = 298) and DEG (*FKBP5* expression and asthma, *N* = 298) analysis pre- and post-adjustment for methylation at the CpG (labeled DEG, unadj and *DEG, adj*).

**Panel C** UCSC Genome Browser view of the *FKBP5* locus, indicating locations of cg03546163 (pcHiC) and cg23416081 (5 kb of TSS) showing interaction between the GeneHancer regulatory elements at these two regions. Publicly available data from tracks displayed includes location of exonic and intronic gene regions from the UCSC gene annotation; regulatory elements, genes and their interactions from GeneHancer, in detailed and clustered views; chromHMM tracks from Roadmap; transcription factor CHIP-seq from ENCODE; and DNAse hypersensitivity density signal from ENCODE for CD20 + B-cells, CD14+ monocytes, fibroblasts and naïve B-cells. Source data are provided as a Source Data file.

between methylation and asthma severity measured by CASI score ($\beta = -0.393$, $p = 8.17 \times 10^{-4}$), than between methylation and ICS usage ($\beta = -0.314$, $p = 0.014$).

## Discussion

In this report, we focus on RNASeq data in nasal epithelium tissue from 536 subjects representing the African Diaspora to define transcriptomic profiles related to asthma. We then integrate these transcript data with DNA methylation also in the same tissue with the goal of identifying mechanisms of dysregulation that underlie molecular subtypes or endotypes of asthma. Limitations in our study include the inability to differentiate gene expression profiles of atopy from asthma given the high prevalence of atopy in our cases and controls, the restriction of methylation data to only the US-based recruitment sites, and our inability to investigate environmental risk factors and social determinants of health in asthma. Despite these limitations, we are uniquely positioned to investigate asthma multi-omics in populations that are historically under-represented in genomics research but bear a disproportionate burden of the disease and disease severity. Overall, we found *N* = 389 differentially expressed genes, and 16 differentially expressed modules that are associated with current asthma adjusting for differences by site and ancestry. We identify strong signatures related to wound healing and drug response at single-gene and network-based levels that may have identified additional endotypes for asthma with potential implications for targeted therapy in the future.

The most significantly differentially expressed gene in asthma cases from CAAPA was *FN1* encoding fibronectin. Despite some between-site differences in overall expression of this gene, there was a consistent lower expression of *FN1* in nasal epithelium in asthma cases compared to controls, and this differential expression of *FN1* was

replicated in data from Tsai et al.[26]. Fibronectin is an adhesion protein. Increased deposition of FN1, along with fibrillar collagen proteins (COL3A and COL4A) in the extracellular matrix (ECM) and sub-epithelial space of airways, results in airway wall thickening and airflow obstruction, thereby altering the structural properties of the airways and the functional properties of airway cells in asthma[34]. In airway epithelial cells (AEC) from children with asthma compared to healthy atopic and nonatopic controls, *FN1* was the only ECM component whose expression was significantly *lower* in AECs from asthma cases[42]. Furthermore, wound healing models using AECs showed that a reduced capacity of AECs to secrete *FN1* contributes to dysregulated AEC repair and impaired wound healing[42].

There are several additional lines of support related to down regulation of genes resulting in impaired wound healing as a prominent feature of asthma risk in CAAPA. First, pathway analysis of the full set of significant DEGs identified *TGFβ1* as an upstream regulator for a network of genes including *FN1*, *COL3A* and *COL4A*. Transforming growth factor, encoded by the *TGFβ1* gene, increases the deposition of ECM proteins, including FN1, and stimulates *FN1* expression in airway epithelial cells[33]. In our study, the expression of all three genes were lower in asthma cases, consistent with impaired wound healing. Second, using a systems biology approach, we identified three modules of co-expressed genes down-regulated in airway cells from the asthma cases. The hub genes from these modules, *FN1* (M5), *ERBB2* (M8), and *ACVR1B* (M23), are associated with wound healing or airway remodeling. *ERBB2* encodes a member of the epidermal growth factor (EGF) receptor family of receptor tyrosine kinases that play a key role in epithelial differentiation, proliferation, and repair[36]. Wound models have shown lower *ERBB2* activation in freshly brushed isolated human AECs from patients with asthma and diminished wound closure and cell proliferation compared to AECs from healthy controls[36]. *ACVR1B*

encodes an activin A type IB receptor for the cytokine activin A which is closely related to *TGFβ1*[43]. In the network of activin A and TGFβ1 regulation, activin A is a potential modulator of airway remodeling[43].

The second axis of dysregulation in asthma cases was a network of upregulated genes that reflect the canonical Th2 pathway. The strongest evidence is seen in the two most differentially expressed modules M6 and M2, which have Th2-related hub genes (*CEACAM5* and *CPA3*, respectively); both are upregulated in the asthma cases. *CEACAM5* is an IL-13-regulated epithelial gene that is upregulated in severe asthma and associated with increased asthma exacerbations[44–46]. It has also been previously identified as a hub gene in a limited microarray-based analysis in 42 cases and 28 controls[47]. Carboxypeptidase A3 (CPA3) is a mast cell (MC) protease and there is a well-known switching of MCs expressing tryptase only (MC$_T$) to MCs expressing tryptase, chymase and CPA3 (MC$_{TC}$) with a dominant expansion of the latter in airways of asthma cases and more specifically in the context of severe asthma[48]. MCs are key effector cells in asthma that are involved in both early and late phase allergic responses[49], and they are most notably involved in the Th2-high endotype of asthma[49]. Additionally, *VSIG4* is the third strongest DEG in our study. It encodes V-set and Ig domain-containing 4 and is an inhibitory ligand on antigen presenting cells thereby regulating T cell responses: macrophages expressing *VSIG4* play a role in inhibiting T cell proliferation and cytokine production[50].

The third axis implicates dysregulated genes that could reflect drug responsiveness. The top 15 DEGs included *PTCHD4*, which was associated with airway disease and identified in a GWAS for oral corticosteroid use[51]; *SPTBN1*, which was identified in a GWAS for leukotriene modifier response in asthma[52]; and *FKBP5*, which plays a role in response to inhaled corticosteroid response as described below. The most significant upstream regulator identified by IPA was the corticosteroid dexamethasone – an oral corticosteroid used in the treatment of asthma, and the 6th strongest upstream regulator was fluticasone propionate – a commonly used steroid treatment for allergic rhinitis in nasal spray formulations. Importantly, we ruled out that the 188 gene targets of these two drugs are directly related to ICS medication use in the asthma cases; 89 genes were targets to both drugs.

The *FKBP5* gene encodes FKBP prolyl isomerase 5 (FKBP5), a cochaperone modulating glucocorticoid receptor (GR) activity associated with an inflammatory response[40]. It was previously shown to be differentially expressed after dexamethasone treatment[41]. Furthermore, a prior study suggested tightly regulated epigenetic control of the expression of genes that modulate GR responsiveness[40]. Here, we identified *FKBP5* in the 15 most differentially expressed genes for asthma and also show epigenetic variation at two interacting regulatory regions possibly accounting for the differential expression. We also observed the largest estimated effect size for this gene was within the Brazil site, which has the most severe asthma cases with greatest proportion of asthma cases on moderate to high ICS use. This observation may be reflective of high *FKBP5* gene expression in these subjects resulting in a decreased responsiveness to ICS, and a consequent escalation of ICS dose. This hypothesis aligns with the requirement of higher doses of ICS in this group of moderate to severe asthma cases from Brazil. By the time the patients were enrolled in the cohort they had difficult-to-treat asthma, but in many cases the disease has been controlled on medium-high doses of ICS combined with formoterol, a long-acting beta 2 agonist bronchodilator (LABA). The severity of asthma could also be related to a long journey to proper treatment, by patients who have suffered from recurrent asthma attacks and symptoms before having access to proper management with free ICS-LABA. Overall, the expression of *FKBP5* was higher in the asthma cases in all other sites, except for those recruited in Denver. In comparison with other CAAPA sites, Denver site participants had the lowest asthma severity indicated by having the lowest mean CASI score (2.81) and highest mean FEV1 (99.5%) predicted in asthma cases (Supplementary

Table S1). The Denver site's low mean CASI score is consistent with mild asthma (CASI ≤ 3)[53,54]; in comparison, the Brazil site's mean CASI score of 8 was the highest for all sites, and has been associated with severe asthma[53]. The Denver site was also distinguished by being 100% pediatric enrollment, differing from the other CAAPA sites with 48.5–100% adult participant enrollment (Brazil site was 100% adult participants). These observations support the relevance of *FKBP5* expression in more severe and persistent disease in CAAPA.

Our integrative multi-omics analyses revealed an important set of genes and co-expressed gene networks with relevant mechanistic roles in asthma that are differentially expressed in the nasal epithelium of asthma cases of African ancestry in CAAPA. This work reveals dysregulation of three axes – increased Th2 inflammation, decreased capacity for wound healing, and impaired drug response. Each is associated with risk for asthma and there is correlation between the axes themselves, but the impact of dysregulation on multiple axes bears a cumulative risk with an OR of 4.95 (95% CI = 3.09–7.93). Networks of genes implicating Th2 inflammation are now well documented, but our findings with respect to gene networks related to wound healing and drug response are not documented, and our findings may be implicating additional endotypes of asthma beyond the well-known type 2 high vs. type 2 low[29]. To date, effective choice of monoclonal antibodies in the management of severe asthma is dependent on the clinical and inflammatory profile of the patient that relates to known endotypes[29]. While future work will be needed to validate these additional axes of dysregulation for drug response and airway remodeling identified in CAAPA and determine if these transcriptomic signatures may be related to ancestry, our findings from this understudied ancestry group that bears significant health disparities in asthma offer the potential to expand our understanding of clinical heterogeneity in disease and treatment response.

## Methods

### Study subjects
Study subjects included African ancestry individuals with no history of COPD, emphysema or chronic bronchitis. CAAPA included adult (aged 18–89) and pediatric (children aged 8–12 and adolescents aged 13–17) individuals. Other exclusion criteria included: pregnancy, lung transplant, kyphoscoliosis, sarcoidosis, bronchiopulmonary dysplasia, cystic fibrosis, bronchiectasis, rheumatoid arthritis, Crohn's disease, psoriasis, lung carcinoma, ciliary dyskinesia, lupus and active tuberculosis. Recruitment occurred at 3 non-US sites (Nigeria, Barbados and Salvador, Brazil) and four US sites (Denver, Baltimore, Washington DC and Chicago). Study subjects were asked to self-identify as African, African American, African Caribbean, African Brazilian or African-Other. Cases were first defined as subjects with 'ever' asthma confirmed by a physician (response = yes to the two questions: (1) Have you ever had asthma?; and (2) Was it confirmed by a doctor?). The final set of cases was further restricted to the subset of individuals with 'current' asthma described below. Controls were defined as subjects with no history of asthma (response = no to the question: (1) Have you ever had asthma?). All samples used for this study were obtained following written informed consent from participants. The University of Colorado (IRB#: 17-1807), Johns Hopkins University (IRB00179053), University of Chicago (IRB18-0466-CR001), National Institutes of Health (IRB#: P184385), University of West Indies (IRB#: 190604-A), University of Bahia (IRB#: 3.302.487) and University of Ibadan Institutional Review Boards approved the conduct of this study (IRB18-0840).

### Sample collection
Nasal columnar epithelial cells from the posterior surface of the inferior turbinate were collected using cytology brushes and standardized protocols[30,55]. After collection, the brush was immediately submerged in Buffer RLT Plus (Qiagen Inc., Valencia, CA, USA) to lyse the

cells and stored until DNA/RNA were extracted. Prior to any processing, a small sample of cells was smeared on a glass slide followed by fixation and H&E staining for quality control. Only samples with ≥80% ciliated epithelial cells visualized from slides were retained. DNA and RNA were extracted from the same nasal sample for multi-omics analysis.

## Questionnaires & phenotype data

Study data were collected and managed using REDCap® electronic data capture tools hosted at Yale University[56] to record health questionnaires from the subjects at all 7 sites. These included informed consent, recruitment forms, respiratory health questionnaires, an asthma severity questionnaire, pulmonary function tests data, complete blood count (CBC) with differentials, vitals collected at the time of visit, medications and physical examination information and date and time nasal and blood samples were collected from the patient. Whole blood collected in BD Vacutainer® EDTA tubes was sent to the clinical laboratory at each site to perform a complete blood count (CBC) with differentials. Serum samples were sent to the Johns Hopkins University School of Medicine Reference Laboratory for Dermatology, Allergy and Clinical Immunology (DACI) for total serum IgE (tIgE) and multi-allergen (phadiatop) measurements. Spirometry was conducted according to the ATS guidelines using a hand-held KoKo DigDoser (Louisville, CO).

## Measurement of atopy, asthma severity, medication use and current asthma

Atopy was defined on the basis of the phadiotop and total serum IgE: if phadiotop was ≥0.36 PAU or IgE was >100KU/L the subject was defined to have atopy. The Composite Asthma Severity Index (CASI) questionnaire[57] was administered to all asthma cases at each of the recruitment sites. CASI takes into consideration medication use and the corresponding treatment level, in determining asthma severity[57]. The scoring guidelines for a treatment component of the CASI test were adapted from the Expert Panel Report 3 (EPR3) asthma guidelines for determining the treatment categories associated with the different levels of medication and a list of medications used for systemic corticosteroid bursts during asthma exacerbations[54,58]. The CASI questionnaire was used to subset asthma cases to those with current asthma given the importance of current disease status for dynamic omics signatures captured in RNA and DNA methylation. Current asthma status was defined as CASI ≥ 1 in asthma cases. Finally, the CASI questionnaire was used to define two medication use groups: no treatment or albuterol as needed (score 0-1 on question 4b) and low to high dose inhaled corticosteroid use (score 2–5 on question 4b). Additionally, at the time of nasal epithelium sample collection, individuals were asked if there were on nasal steroids, and if they were able to withhold usage for 5 days prior to sampling: there were 68 cases on nasal steroids, of whom 32 did not withhold usage.

## Clinical characteristics of study subjects

Clinical characteristics of study subjects used for generating multiomic data sets are summarized in Supplementary Data 1. Adults ranged in age from 18 to 89, and pediatric subjects ranged in age from 8 to 17. Additional information on sample distributions by phenotypes and their associations is also shown in Supplementary Data 1. For each quantitative phenotype (CASI, tIgE, eosinophil count, phadiotop, FEV1, FEV1/FVC), we tested for differences between cases and controls in the full group using a linear model adjusting for age, sex, the first two PCs plus recruitment site site (Supplementary Data 1, Full Group). Additionally, because Chicago was the only site to recruit both adults and children, we used this as the reference group to compare phenotypes across sites. In those analyses, we used similar linear models for the quantitative phenotypes and a logistic model for medication use,

stratifying by case and control status and adjusting for age, sex, and the first two PCs.

## Omics data generation

Genotyping was performed on DNA extracted from blood clots from CPT tubes on samples from all 7 recruitment sites. Samples with DNA Integrity Number (DIN) ≥6 were run on Illumina's Infinium® Multi-Ethnic Global BeadChip (MEGA). Following the Illumina Infinium protocol, idat files were generated and used to extract the genotype calls, perform data QC and downstream analysis. Genotype data was used to derive ancestry principal components (PCs). Genotyping was performed on samples from all 7 recruitment sites.

RNA samples from nasal airway epithelial cells collected across all 7 recruitment sites and that passed all the laboratory QC thresholds (at least 80% columnar cells per nasal slide assessment, nanodrop and Qubit concentration ≥30 ng/ul in 20 uL volume (a total amount of 600 ng), RIN ≥ 6 and 260/280 between 1.7 and 2.2) were sent for RNA sequencing to Psomagen. RNASeq batches were balanced with respect to site, asthma status, sex and age (adult *vs.* child) to minimize confounding. The Illumina TruSeq Stranded Total RNA with Ribo-Zero kit was used to prepare libraries, depleting the ribosomal RNA. RNA sequencing (RNAseq) was performed on the NovaSeq 6000 using 150 bp paired end reads and yielding at least a total of 80 million paired end reads per sample. RNASeq was performed on samples from all 7 recruitment sites.

DNA methylation (DNAm) quantification was performed using Illumina's Infinium MethylationEPIC array® using genomic DNA from nasal airway epithelial cells collected across the 4 US recruitment sites that passed all the laboratory QC thresholds (at least 80% columnar cells per nasal slide assessment, nanodrop and Qubit concentration ≥15 ng/ul in 50 ul (a total amount of 750 ng), DIN ≥ 6 and 260/280 between 1.4 and 2.15). Bisulfite conversion was performed using the EZ DNA Methylation™ kit (Zymo Research) and DNAm quantification was performed using the Illumina Infinium protocol. Idat files generated from the Infinium protocol were used to perform the data QC and downstream analysis. Methylation chips were randomized with respect to site, asthma status, sex and age (adult *vs.* child) to minimize confounding.

## Quality control (QC) and preprocessing

Quality control procedures were performed on genotype data to exclude any samples or variants that had missingness >3%; standard QC steps included: sex verification, heterozygosity checks and identity-by-descent (IBD) to look for any unexpected relatedness. Two individuals failed on call rate and four individuals were identified as sex mismatches in one or more omics data sets (dropped from all datasets). Thirteen individuals showed cryptic relatedness, as being part of a parent offspring pair, full sibs or half sibs pairs based on the IBD estimates; one independent subject was selected prioritizing case status from each relationship resulting in 7 individuals being dropped from all datasets. One sample was identified as a sample swap, duplicating another individual, and was dropped. A total of 14 samples were excluded and further ancestry analysis was limited to 673 individuals from all seven sites.

To perform RNAseq QC pre-alignment, QC, adapter trimming, and alignment of reads to GRCh38 were performed using FastQC [http://www.bioinformatics.babraham.ac.uk/projects/fastqc/], Picard tools [http://broadinstitute.github.io/picard], BBDUK [sourceforge.net/projects/bbmap/], and HISAT2[59], respectively. Raw counts were generated by CoCo[60]. Four samples were excluded due to sex mismatches, 3 samples were excluded due to unexpected relatedness, 4 samples were excluded due to failure of library preparation, and 7 samples were excluded due a high percentage of ribosomal RNA. After filtering asthma cases to include only current asthmatics, and

available genotype data for ancestry PCs, analysis was limited to 536 individuals from all seven sites.

DNAm data QC was tested using the minfi R-package[61]. Samples were excluded based on the following metrics: 12 with mean of methylated and unmethylated signal <10.5; four with discordant sex. CpG sites were filtered as follows: $N = 953$ probes that had fewer calls detected (detection $p > 0.01$) in more than 20% of the samples; $N = 30,247$ probes that were in close proximity to SNPs (at the CpG site or in the single-base extension site for the array probe), yielding a total of 834,663 CpGs that were normalized using the quantile option for downstream analysis. After filtering asthma cases to include only current asthmatics, and available genotype data for ancestry PCs, analysis was limited to 331 individuals from four US-based recruitment sites: Baltimore, Chicago, Denver and Washington DC.

## Quantification of principal components and global sample ancestry

We used KING[62] to estimate relatedness and build a kinship matrix including all subjects from all the sites. SNPs with >5% missingness, <1% minor allele frequency and hardy Weinberg $p < 1 \times 10^{-6}$ were discarded. SNPs underwent linkage disequilibrium-pruning (removal of SNPs with an $R^2$-value > 0.1 within every 50 SNP window) and PCA analyses were performed using a CAAPA-only dataset comprised of 673 samples and 512,925 SNPs. PCA was performed allowing for observed kinship using PC-AiR as implemented in the GENESIS R package[63]. Additionally, PCA was also performed including reference populations from the 1000 Genomes Project (85 Utah residents with Northern and Western European ancestry CEU (EUR), 88 Yoruba samples from Ibadan, Nigeria YRI (AFR) and 43 Native Americans selected from Mao et al.[64] (AMR)) on 219,832 autosomal SNPs obtained after the merge with reference data. The elbow in the scree plot was used to identify the top two PCs (PC1 and PC2) as covariates for ancestry adjustment, Supplementary Fig 1.

To estimate global ancestry proportions, we first implemented cross-validation using ADMIXTURE[65] to determine the number of reference populations (K) with $K = 1-5$, to infer the optimal number of ancestral reference groups needed. $K = 3$ accounted for the lowest cross-validation error. CAAPA samples were merged with 3 reference populations and with the set of 219,832 autosomal SNPs obtained after the merge, and using K = 3, we performed global ancestry estimation using ADMIXTURE and plotted the admixture estimates using the PONG visualization tool[66] as shown in Supplementary Fig 1.

## Differential gene expression analysis

Mean normalized counts were generated by DESeq2[67]. Genes were first filtered across all sites combined to only include those with DESeq2 mean normalized count $\geq 20$ and counts per million (CPM) > 0 in the percentage of samples corresponding to the proportion of asthma cases to never-asthma controls (e.g. in the full sample there were 47.20% cases, and we required CPM > 0 in 47.20% of the total sample); this was done to ensure expression variability was present in both cases and controls. Following filtering, 21,831, 21,789 and 21,887 genes were available for analysis in the full dataset, adult-only dataset and pediatric-only dataset, respectively. Differential gene expression analyses for case-control comparisons were performed on all subjects, and stratifying into adult and pediatric groups, plus stratifying subjects by site. All site analyses were performed on the same set of filtered genes generated for the analysis of all subjects above. Differential analysis in limma[68] and edgeR[69] was performed. Counts were transformed to $\log_2$ (CPM) using voom[68] and a linear model was fit adjusting for relevant covariates: asthma status, age, sex, library preparation batch, site, RNA integrity number (RIN), GC content, and ancestry PCs 1 and 2. Sites were not included as a covariate in these site specific analyses. SVA[70] was used to generate surrogate variables (SVs) for each stratified analysis and significant SVs were added to the model. Analysis was also

performed in the subset of $N = 253$ asthma cases for medication use including the same covariates, and SVs derived on only these subjects.

## Ingenuity pathway analysis

Differentially expressed genes with $q < 0.05$ in the analysis of all subjects were selected as input for Ingenuity pathway analysis (IPA)[71]. IPA upstream regulator analysis was utilized to identify molecules upstream of the selected DEGs that could potentially explain the observed expression differences between case and controls. Activation z-score (a measure of consistency between up/down gene regulation pattern and activation/inhibition pattern given by the IPA knowledgebase network) and p-value of overlap (a measure of significance of enrichment of regulated genes in the dataset, agnostic to direction) were generated for each upstream regulator, and gene targets of each regulator were obtained from the upstream regulator table. P-value of overlap measures of significance of enrichment of regulated genes in the dataset given a regulator, agnostic to direction; all upstream regulators have a generated $p$-value. IPA predicts the activation state of the regulator by assessing the consistency of direction of gene expression of gene targets in the dataset with activation/inhibition patterns given by the IPA knowledgebase relative to a random pattern. Activation z-score was calculated for regulators given that the direction of regulation is well defined based on literature findings in the IPA Knowledgebase and the underlying null model is appropriate; therefore, not all upstream regulators have an activation z-score.

## Gene expression module analysis

Differentially expressed genes with $q < 0.15$ in the analysis of all subjects were selected for weighted gene coexpression network analysis (WGCNA)[72]. This more liberal significance threshold was used to cast a wide net for a systems biology analysis. WGCNA was performed on 1326 genes using mean normalized counts described above corrected for library preparation, batch, sex, RIN, and GC content with the following parameters: power was selected for which the scale-free topology fit index reached 0.9, network type = "signed", TOM type = "signed", deepSplit = 3, min module size = 15, max block size = 8000. Gene module differential expression analysis with average expression across genes in a module as the outcome was performed using limma according to the analysis pipeline described above for gene differential expression comparing current asthma cases to never-asthmatic controls without the use of SVs, but otherwise adjusting for the covariates age, sex, library preparation batch, site, RNA integrity number (RIN), GC content, and ancestry PCs 1 and 2. Multiple testing correction was performed using the Benjamini-Hochberg method and an FDR cutoff <0.05 was considered significant.

Significantly differentially expressed gene modules were visualized using cytoscape[73] and STRINGdb[74]. Module expression was calculated as the mean expression of genes, using voom transformed counts corrected for sex, library preparation batch, RIN, and GC content using limma removeBatchEffect(), assigned to a module and STRING protein-protein interaction networks were retrieved for differentially expressed modules. Interactions with stringdb score >0.15 were considered significant and unconnected genes were omitted. Gene hubs for each module were determined as the gene with the highest connectivity in the module using the chooseTopHubInEachModule() WGCNA function or the gene with the highest connectivity in the module STRING network.

## Multi-module analysis

To evaluate the cumulative risk for asthma across the three axes of dysregulation, we selected the most significant module based on q-value for Th2 inflammation (M6, *CEACAM5*), wound repair (M5, *FN1*), and drug response (M4, *NCALD*). Covariate effects (age, sex, site, RIN, GC content, library preparation batch, ancestry PC1 and PC2) were

regressed out from module average expression. Each module was dichotomized with values of 0 and 1 at the median considering direction of effect of module association with asthma: for M6 where higher module expression is associated with asthma risk, the upper 50th percentile was labeled as = 1, and lower 50th percentile labeled as 0; for M4 and M5 where lower module expression is associated with asthma risk, lower 50th percentile was labeled as = 1 and upper 50th percentile was labeled as = 0. We then calculated the sum of these three binary covariates (the "number of modules" a sample is in the risk side of the median dichotomy) as a covariate of interest and used logistic regression models to investigate the association between this number and the fraction of asthma cases observed in our data. The number of modules was fit as a numeric term, but we also tested formally for departures from additivity by comparing this model in a likelihood ratio test to a logistic model with number of modules as a factor, allowing for an extra 2 degrees of freedom.

### DNA Methylation and multi-omics analysis

For the set of $N = 389$ asthma DEGs, eQTM analysis was performed by fitting a linear model for each gene-CpG pair to test the association between gene expression and *cis*-CpG methylation using the Matrix eQTL R package[75], testing all CpGs within 1 megabase of the nearest end of the gene. As we are modeling gene expression as the outcome, all the covariates used for differential expression analysis were included: case-control status, age, sex, library prep batch, site, RIN score, GC content, ancestry PC1 and PC2. As generation of SVs using SVA would need to be performed for each CpG, we instead generated PEER factors[76] using the peer package in R with all covariates listed above and default parameters (Alpha_Prior_A = 0.001, Alpha_Prior_B = 0.01, Eps_Prior_A = 0.1, Eps_Prior_B = 10, Max_Iteration = 1000, Tol = 0.001). We included 60 PEER factors in our eQTM models. eQTM analysis was run on a total of 298 subjects where both RNASeq and DNAm data were available. CpGs were selected if they mapped within 5 kb of a gene transcription start site (upstream or downstream) or if they were annotated by promoter-capture HiC in bronchial epithelial cells[77] to lie in putative enhancer regions for these 389 genes. There were 8418 eQTM tests performed for gene-CpG pairs and significance for the eQTMs was defined as eQTM $p < 0.05$.

For the subset of CpGs identified as significant eQTMs, we tested for differential methylation (DMC analysis) using standard linear modeling approaches implemented through the limma R package[68], including age, sex, site, plate, ancestry PC1 and PC2 and 12 latent factors estimated from CBCV-CorrConf R package[78] included to adjust for additional unmeasured confounders such as cell composition differences. Differential methylation analysis was run between 149 cases and 182 controls for 915 CpGs. Significance was defined at the Bonferroni threshold of $p < 0.05/915$. For the subset of CpGs that were identified to be eQTMs and DMCs, and where there were multiple eQTMs per gene, a joint model was run to evaluate independence of the multiple CpGs.

Finally, in the 298 subjects that had both DNAm and RNAseq, conditional analysis was performed for the CpG's identified as both eQTMs and DMCs to understand the effect of DNAm on the relationship between gene expression and asthma, i.e. was there a gene expression association with asthma after adjustment for the relevant DMC. The following models were run: DEG,$_{unadj}$ testing for differential gene expression including all the original covariates (asthma status, age, library preparation batch, site, RNA integrity number (RIN), GC content, ancestry PCs 1 and 2, 23 SVs), and DEG,$_{adj}$ testing for differential gene expression including all the original covariates and also including methylation at the CpG of relevance for the gene (i.e. peak eQTM for the gene). We examined the change in effect size and significance of the DEG between the DEG,$_{unadj}$ and DEG,$_{adj}$ models.

### Replication

For the 21,831 genes tested in the DEG analysis of all subjects, we performed an exhaustive search for replication in a meta-analysis study of airway epithelium gene expression in asthma[26]. Briefly, Tsai et al. performed a meta-analysis of eight independent gene expression studies including both nasal and bronchial epithelium tissue. Full results from the meta-analysis were obtained from these authors and were compared to the CAAPA results by matching genes on Ensembl ID. Ensembl IDs were retrieved for the meta-analysis gene symbols from Ensembl Release 109 homo sapiens GRCh38 using pyensembl [https://github.com/openvax/pyensembl]. Each gene symbol in an observation was queried individually for matching Ensembl IDs and matched to Ensembl IDs in CAAPA. Where multiple observations in the meta-analysis matched an Ensembl ID in CAAPA, the observation with the highest number of studies (k) in the meta-analysis was selected. Enrichment of CAAPA DEGs in the meta-analysis DEGs was tested using a hypergeometric test. The total number of genes tested was determined as the number of unique Ensemble IDs retrieved in the meta-analysis full results that matched to Ensembl IDs included in the 21,831 genes tested in CAAPA.

### Reporting summary

Further information on research design is available in the Nature Portfolio Reporting Summary linked to this article.

### Data availability

The RNASeq data generated in this study have been deposited in the GEO database under accession code GSE240567 [https://www.ncbi.nlm.nih.gov/geo/query/acc.cgi?acc=GSM7701971]. The Methylation data generated in this study have been deposited in the GEO database under accession code GSE250513. The genotype data generated in this study have been deposited in dbGAP database under accession code phs001123. Meta-analysis result from Tsai et al. are available in Supplementary Data 2. Source data are provided with this paper for all figures. Source data are provided with this paper.

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

## Acknowledgements

We thank the numerous health care providers, and community clinics and co-investigators who assisted in the phenotyping and collection of samples, and the CAAPA participants for generously donating time and samples. We gratefully acknowledge the contributions of clinical recruiters and technicians from each recruitment site: Allison Schiltz, Alan Franklin, Elizabeth Davidson (Denver); Amy Bentley, Lin Lei, Carolyn Allen, Paule Joseph, Hermon Feron (Washington DC); Carly Jackson and Robert Stanaker (Chicago); Laura Grammer (Baltimore); Dr. Ayobami Bakare, Dayo Adepoju, Samuel Adekunle, Tope Ibigbami, Abayomi Odetunde, Sunkanmi Owolade, Joseph Eleyinmi (Nigeria); Pissamai Maul, Trevor Maul, Desiree Walcott, Andre Greenidge (Barbados); Luane Marques Mello, Gabriela Pimentel Pinheiro das Chagas, Cinthia Vila Nova Santana, Débora Inácio, Laila Trindade, Givaneide Lima, Tamires Carneiro, Candace Andrade, Helena Teixeira, Ryan Costa (Brazil). We are grateful for the support from the international state governments and universities from Barbados, Brazil and Nigeria who made this work possible. We acknowledge the support from James Kiley and Weiniu Gan at the National Heart, Lung and Blood Institute. Funding for this study was provided by National Institutes of Health (NIH) R01HL104608. We thank Christopher Arehart, Iain Konigsberg and Chris McKennan for their support and input on bioinformatic pipelines leveraged in this work.

## Author contributions

Data analysis was performed by C.C., A.M. and W.L. Data analysis, and writing was carried out by B.S., M.P.B., S.C., M.C., R.K.J., K.K., E.E.T., G.S., M.D., S.N.P.K. and M.S.C. Scientific leadership and data acquisition were provided by H.W., E.T.N., B.L.G., G.A.A., O.S., A.G.F., N.N.H., C.N.R, R.C.L. and E.K. Data analysis, scientific leadership and writing were performed by I.R., A.H.L., C.O., M.A.T., K.C.B., R.A.M. and M.C.A. Scientific leadership and writing were provided by THB,APD,AAC,IVY,COO and CAF.

## Competing interests

K.C.B. declares Royalties from UpToDate. The remaining authors declare no competing interests.

## Additional information

Brooke Szczesny[1,29], Meher Preethi Boorgula[2,29], Sameer Chavan [3], Monica Campbell[3], Randi K. Johnson [4,5], Kai Kammers[6], Emma E. Thompson [7], Madison S. Cox[7], Gautam Shankar[1], Corey Cox [2], Andréanne Morin [6], Wendy Lorizio[1], Michelle Daya[2], Samir N. P. Kelada [8,9], Terri H. Beaty [10], Ayo P. Doumatey[11], Alvaro A. Cruz [12], Harold Watson[13], Edward T. Naureckas[14], B. Louise Giles[15], Ganiyu A. Arinola[16], Olumide Sogaolu[17], Adegoke G. Falade[18], Nadia N. Hansel[1], Ivana V. Yang[19], Christopher O. Olopade[20], Charles N. Rotimi [11], R. Clive Landis[21], Camila A. Figueiredo [22,23], Matthew C. Altman [24,25], Eimear Kenny[26], Ingo Ruczinski[27], Andrew H. Liu[28], Carole Ober [6], Margaret A. Taub[27], Kathleen C. Barnes[2,30] ✉ & Rasika A. Mathias [1,30] ✉

[1]Department of Medicine, Johns Hopkins University, Baltimore, MD, USA. [2]Department of Medicine, University of Colorado Denver, Anschutz Medical Campus, Aurora, CO, USA. [3]Department of Biomedical Informatics, University of Colorado Anschutz Medical Campus, Aurora, CO, USA. [4]Department of Epidemiology, Colorado School of Public Health, Aurora, CO, USA. [5]Quantitative Sciences Division, Department of Oncology, Johns Hopkins University School of Medicine, Baltimore, MD, USA. [6]Departments of Human Genetics, University of Chicago, Chicago, IL, USA. [7]Division of Allergy and Infectious Diseases, Dept of Medicine, University of Washington, Seattle, WA, USA. [8]Department of Genetics, University of North Carolina, Chapel Hill, NC, USA. [9]Marsico Lung Institute, University of North Carolina, Chapel Hill, NC, USA. [10]Department of Epidemiology, Johns Hopkins Bloomberg School of Public Health, Baltimore, MD, USA. [11]Center for Research on Genomics and Global Health, National Human Genome Research Institute, National Institutes of Health, Bethesda, MD, USA. [12]Fundacao ProAR and Federal University of Bahia, Salvador, Bahia, Brazil. [13]Faculty of Medical Sciences, The University of the West Indies, Queen Elizabeth Hospital, St. Michael, Bridgetown, Barbados. [14]Departments of Medicine, University of Chicago, Chicago, IL, USA. [15]Departments of Pediatrics, University of Chicago, Chicago, IL, USA. [16]Department of Immunology, College of Medicine, University of Ibadan, Ibadan, Nigeria. [17]Department of Medicine, College of Medicine, University of Ibadan, Ibadan, Nigeria. [18]Department of Pediatrics, University of Ibadan, and University College Hospital, Ibadan, Nigeria. [19]Departments of Biomedical Informatics and Medicine, University of Colorado Denver, Anschutz Medical Campus, Aurora, CO, USA. [20]Department of Medicine, University of Chicago, Chicago, IL, USA. [21]Edmund Cohen Laboratory for Vascular Research, George Alleyne Chronic Disease Research Centre, Caribbean Institute for Health Research, The University of the West Indies, Cave Hill Campus, Wanstead, Barbados. [22]Federal University of Bahia and Funda. Program for Control of Asthma in Bahia (ProAR), Salvador, Brazil. [23]Instituto de Ciências de Saúde, Universidade Federal da Bahia, Salvador, Brazil. [24]Systems Immunology Program, Benaroya Research Institute, Seattle, WA 98101, USA. [25]Division of Allergy and Infectious Diseases, Department of Medicine, University of Washington, Seattle, WA 98109, USA. [26]Center for Genomic Health, Icahn School of Medicine at Mount Sinai, New York, NY, USA. [27]Department of Biostatistics, Johns Hopkins Bloomberg School of Public Health, Baltimore, MD, USA. [28]Department of Pediatrics, Childrens Hospital Colorado and University of Colorado Denver, Anschutz Medical Campus, Aurora, CO, USA. [29]These authors contributed equally: Brooke Szczesny, Meher Preethi Boorgula. [30]These authors jointly supervised this work: Kathleen C. Barnes, Rasika A. Mathias. ✉e-mail: kathleen.barnes@cuanschutz.edu; rmathias@jhmi.edu

