## [Peer Review File · Nature Communications]

Multi-omics in nasal epithelium reveals three axes of dysregulation for asthma risk in the African Diaspora populationsREVIEWER COMMENTS

Reviewer #1 (Remarks to the Author):

- The article is well written. However, I have the following comments
- The study repeatedly use asthmatic compared with never-asthmatic controls or asthma compared with controls. The controls are primary based on family history and it not clear if clear never-asthmatic controls are correct. Does it mean no other allergic diseases as comorbidities as well.
- It is not clear how the genes from multi-omics integration identified.
- It is not clear why methylation was taken as mediator (since it is know it regulates gene expression)
- The study lacks multiple testing correction.
- The study (including the title) presented as African diaspora. However, the samples include diaspora and non-diaspora. It is not scientifically accurate to characterize population from different regions as diaspora. I suggest removing diaspora from the title.
- The 4th paragraph of the introduction should not be part of the the introduction. It is about methods and results.
- The liberal significance threshold was used. However, it is not clear what cut off and why. Otherwise, the analysis is descriptive and subjective.
- For IgE differences between sites, did you look other allergic diseases. This is very important. Ideally, all allergic diseases should be include as covariates.
- Lifestyle, environmental exposure should be included as covariates before comparing sites.
- It is not clear if generating genes based on significant DEGs followed by WGCNA is better or directly running WGCNA is appropriate.. Comparison will be useful.
- It is not clear why the DNA methylation results were conditional on significant DEGs. What about the other way round. To look significant DNMA CpGs and then look for DEGs.
- The replications are not exhaustive and true replication.
- True validation of relevant genes including functional analysis are critically important
- Figure 2C, three modules (m6, m5 and M4) expression are used for plotting case control status. However, the three modules do not clearly separate cases from the controls. Do you check other modules? For example, M3 had high FC = 0.16 instead of using M4 (FC = -0.13). I expect clear separation of the cases and controls in Fig 2C and higher OR in Fig2D.
- Childhood asthma is different from adulthood asthma. So separate analysis should be followed throughout the manuscript.
- It is not clear what second phase of CAAPA refers to.
- There is no plan to submit the data to the public repository like GEO so that others can replicate the study.

Reviewer #2 (Remarks to the Author):

This is a well written paper which addresses nasal epithelial gene expression differences between asthma cases and controls in a population of African ancestry subjects. The main conclusions are that pathways involved in wound healing (and in particular a role for FN1) and drug responsiveness are differentially regulated in cases v controls. Additional data trying to address epigenetic regulation are included using eQTM approaches. Whilst many of the findings reinforce the role for a range of genes for which a role in asthma is already well established (eg many genes involved in Th2 responsiveness) there are fewer data available from this ancestry group, and overall therefore the findings add to the current literature. The methodology is sound.

There are a few points which the authors could consider addressing:

1 Population characteristics. The populations used are relatively small from each site (the smallest being 62 individuals, see supplementary table 1) and of quite diverse ancestry (see supplementary figure 1a). Whilst this has been factored into the analyses presented, how these differences were dealt with probably deserves some further comment; in particular the epigenetic analyses were only performed in USA subjects.

2 Treatment effects. Many of the subjects in addition to having asthma had other features of allergic disease. How many were taking nasal steroids? The table suggests 61% were using 'as required' or no medication (I presume this refers just to asthma).....did this include ICS or just SABAs? This does matter as the conclusions around FKBP5 being upregulated in asthma could merely be due to medication rather than the underlying disease. Indeed, the effect size for FKBP5 was higher in those on regular higher dose steroids.

3 It is reassuring that in the replication analyses for the DGEA 87/353 replicated with for 86 signals this having the same direction of effect in the dataset used (ref 26....note this is bronchial epithelial rather than nasal epithelial but the authors provide support that there is reasonable concordance between the 2 sites in gene expression changes in asthma). The data for FN1 look robust across these data sets. However, if the aim in part was to look for ancestry specific effects, using a replication set which isn't in the same ancestry group may lead to a failure to replicate.

4 In the eQTM analyses it looks as if 8418 gene CpG pairs were examined. I assume though that methylation was studied genome wide, so there must be extensive other data available which are not presented in this paper. Given the number of methylation sites on the current chips including genome wide methylation analyses to explore trans effects would present a challenge without a replication population so I think this is acceptable. 918/8418 gene CpG pairs were considered significant at $p=0.05$ although many of these could be false positives given the low stringency p value used: this probably deserves comment. Only 5 DMCs survived multiple testing correction, but the FN1 locus was not one of these. What do the authors infer about these relationships....do they think they may be causal, or due to the effect of the presence of disease, or due to drug treatment (I note that FKBP5 comes up here)?

Reviewer #3 (Remarks to the Author):

Summary

Szczesny et al obtained RNA-Seq and Methylation data for nasal epithelium samples from 253 asthma cases and 283 never-asthma controls from a cohort called Consortium on Asthma among African-ancestry Populations in the Americas (CAAPA). Analyses included differential gene expression, pathway analysis, gene co-expression network analysis and eQTMs and differential methylation sites for select genes. Some analyses included fewer subjects (eg DNA methylation and RNA-Seq was 298 total). They found various genes implicated in asthma via these analyses, including glucocorticoid responsive genes. A few lines of results are interesting (FN1 as top differentially expressed, upstream

regulators, FKBP5 and methylation results integrated with differential expression) but they are not functionally validated or linked in such a way that provides compelling new insights into asthma or its related processes. Most results presented are then supported by prior literature which strengthens likelihood that results are generalizable while also diminishing novelty of paper given that hardly any findings remain truly novel. Loosely collating the results of analyses into three axes seems to be an attempt to make a cohesive story out of a few lines of research that are not fully pursued. The results may nonetheless interest others in the field who may want to corroborate their own findings or pursue experiments based on the hypotheses generated by the work presented.

Major

- 1) The introduction and discussion bring up "In this second phase of CAAPA," but it is not clear what the first phase was – the GWAS of a prior introduction paragraph? Or, is there something else that is being referred to? How many phases does CAAPA have?
- 2) Referring to three processes as axes implies independence among them that likely does not reflect biological processes given that Th2 inflammation, wound repair and drug responses to steroids and other asthma drugs are all interrelated. More clarity around the limitations implicit in taking a cross sectional view that is displayed by the modules should be provided as the processes are not truly independent.
- 3) Gene expression signatures by their nature are highly variable and dependent on many environmental and internal factors. Thus, while it is impressive that the authors collected samples from people from 7 different geographic locations, having only ~500 people represented is a limitation to establish generalizability. Based on results from prior transcriptomic studies of asthma, there is no clear asthma "signature" and cluster analysis has not yielded highly reproducible groups aside from the Th2 high and low ones. There are many differences across sites in terms of severity, drug use, age, etc, which also confound relationships observed. This needs to be described as a major limitation of the work.
- 4) Table 1 should include participant characteristics given that this is key information for the reader to interpret study design and findings, and not be relegated to the supplement.
- 5) Having DNA methylation only from four sites, all in the US, is another limitation that introduces bias in the results from this modality.
- 6) Multi-module analysis is a bit arbitrary as presented. Were individual modules for the traits in question determined on the basis of single genes? For the WGCNA analysis, the modules are based on groups of genes for which the authors do not say much. Representing each module as a single gene as in Table 1 (or no genes as is the case for some) is not helpful for interpretation. An ontological enrichment analysis of the genes in each module, at least for the top modules, may help with interpretation. Similarly, listing some of the many genes within them can help with interpretation, at least for some top genes that are differentially expressed.
- 7) For testing of 8418 eQTM, the significance threshold ($p < 0.05$) was not corrected for multiple comparisons made. This suggests those results are less reliable than any others.
- 8) Typo line 260: "The effect of methylation was evaluated looking at the change in effect size and significance of the DEG between the DEG_{unadj} and DEG_{unadj} models."
- 9) Replication occurring in airway epithelium is a limitation. Although nasal and airway are similar, they are not the same. This is not mentioned in discussion section.
- 10) Searching for upstream regulators rather than pathways represented by the 389 DEGs is interesting, but how would no genes that are regulators of the DEG ones not be differentially expressed themselves?
- 11) Along those lines, glucocorticoid use and responsiveness is likely different among subjects and impacts the DEGs as suggested by IPA analysis. How do authors explain that no genes regulated by the drugs are differentially expressed yet the upstream regulators are significant? FKBP5 is a canonical marker of glucocorticoid response in vitro, with levels that rise shortly after exposure and are maintained for at least 24 hours. Suggesting that levels of such a gene that vary over short time periods are stable biomarkers is a stretch despite some evidence from a publication that it could be a drug response biomarker. Without knowing the time frame when a person took medications, it is difficult to conclude much.

12) Sentence starting on line 326 mentions 879 differentially expressed genes, in contradiction with number used for IPA

13) Figure 2 panel C does not convey any strong results given the large scatter of points from cases/controls across the 3D space. Choosing a 3D plot to be represented in 2D is usually not helpful as this plot demonstrates. Panel D perhaps makes a better case although there is some tautology in showing that more the groups defined by the differentially expressed genes used to define the groups confer differences in asthma risk.

14) Figure 3 lacks clarity in labels for Panel A (especially for box plots), message conveyed for Panel B, and sources of data for Panel C. For Panel C, please clarify that nothing displayed corresponds to authors findings. Legend mentions general data represented, but what are the many tracks shown and why were they selected?

15) Claims about identification of endotypes and implications for targeted therapy should be removed given that authors did not explore or confirm clusters, nor do they have an appropriate sample size to do so

16) Discussion does not include clear listing of limitations, of which there are many important ones. Instead, the final paragraph is overly optimistic about results being helpful for endotyping, finding new drugs, asthma biologics, etc.

Minor

1) Typo line 149: therapy

REVIEWER #1

(1) The study repeatedly use asthmatic compared with never-asthmatic controls or asthma compared with controls.

Response: We have now used the terms *case* and *control* consistently, defining case as current asthma and control as never asthma only where needed.

Changes:

Abstract: Cases (current asthma, N=253) were compared to controls (never-asthma, N=283) to identify differentially expressed genes (DEGs; q <0.05).

Methods: RNA sequencing and DNA methylation data from the same nasal epithelium samples in cases (individuals with current asthma) and controls (individuals never having asthma) representing 7 locations across the African Diaspora

Restuls: Cases (N=253) with current asthma status and controls with never-asthma status (N=283) were recruited from seven sites including 4 US-based locations and 3 international locations (Table S1).

(2) The controls are primary based on family history and it not clear if clear never-asthmatic controls are correct.

Response: To clarify, the controls are not based on family history, but rather a self-history from questionnaire data. In detail, subjects were defined as controls if their response to the question “Have you ever had asthma?” was ‘no’, i.e. subject has never had asthma. We have edited the methods to use the term ‘self-history’ to avoid confusion with family-history in the main body of the paper. Additionally, we have added much more detail in the Supplementary Methods to minimize any confusion.

Changes:

Methods: Controls were defined as subjects with no self-history of asthma.

**Also see Supplementary methods*

(3) Does it mean no other allergic diseases as comorbidities as well. For IgE differences between sites, did you look other allergic diseases. This is very important. Ideally, all allergic diseases should be include as covariates.

Response: We expect co-occurrence of disease phenotypes within the allergic diathesis.

Asthma	Hay fever	Food Allergy	Prolonged itch	Any co-morbidity
Case	70%	44%	47%	81%
Control	27%	10%	15%	35%

Our classification of asthma is on the basis of a doctor’s diagnosis, in contrast hayfever, food allergy and prolonged itch (a proxy for eczema) is a self-report without a doctor’s confirmation.

Differences in IgE could certainly reflect other allergic conditions, and our controls were not selected to be non-atopic, but only non-asthmatic. We appreciate the reviewer’s comment that the comparison of IgE between site in the controls does not address the issue of allergic phenotypes, and we have repeated the IgE comparison in the controls including the presence of comorbid allergic conditions based on self-report as a covariate.

	Nigeria	Barbados	Washington DC	Baltimore	Chicago	Denver	Brazil	
Total IgE in controls (mean*, 95% CI)	48.1 (26.6-87.0)	21.3 (13.9-32.7)	43.2 (31.2-60.0)	22.1 (12.7-38.7)	20.8 (12.9-33.5)	24.8 (16.3-37.8)	23.9 (14.0-40.6)	Unadjusted
p-value	0.1390	0.7590	0.0368	0.7070	-	0.8240	0.0967	
Total IgE in controls (mean*, 95% CI)	48.1 (26.6-87.0)	21.3 (13.9-32.7)	43.2 (31.2-60.0)	22.1 (12.7-38.7)	20.8 (12.9-33.5)	24.8 (16.3-37.8)	23.9 (14.0-40.6)	Adjusted
p-value	0.1027	0.6746	0.0915	0.8171	-	0.9878	0.1358	

We did not see any major changes: previously the only site with a significant p-value compared to Chicago was Washington DC, and it is no longer significant after adjustment for allergic co-morbidities (p-value old:0.0368 p-value new: 0.0915). We have updated Table S1 to reference this point.

Changes: See Table S1.

(4) It is not clear how the genes from multi-omics integration identified.

Response: We apologize if this was not clear. This is now clarified in the Methods section. DNA Methylation and multi-omics analysis: The multi-omics analysis was limited to the set of N=389 asthma DEGs.

Changes:

Methods: *The multi-omics analysis was limited to the set of N=389 asthma DEGs identified with q-value <0.05.*

(5) It is not clear why methylation was taken as mediator (since it is know it regulates gene expression). It is not clear why the DNA methylation results were conditional on significant DEGs. What about the other way round. To look significant DNMA CpGs and then look for DEGs.

Response: We regret using the term mediation; our intention from this set of conditional models was to see if the association between gene expression and asthma was regulated by methylation. Specifically in the multi-omics analysis where we identified a CpG to be both an eQTM for a DEG and a DMC for asthma, we wanted to test to see if there remained an association between the gene expression and asthma that was independent of methylation effects at the CpG. Hence, the statistical models were set up to determine if *after adjusting for methylation there was an independent effect of gene expression*. We did not run statistical analysis to see if DNAm results were conditional on gene expression, we did however limit any multi-omics analysis involving DNAm to the set of genes that were identified as DEGs.

Changes:

Abstract: *Multi-omic analysis identified FKBP5 as a key contributor to asthma risk, where the association between nasal epithelium gene expression is likely regulated by methylation and is associated with increased use of inhaled corticosteroids.*

Figure 3: Epigenetic mechanism relating gene expression to asthma for FKBP5. Panel A: Scatter plot of methylation (beta) values at cg03546163 vs gene expression (log2 CPM) values for FKBP5 and box plots showing median and interquartile range by asthma case and control status for N=298 individuals.

Panel B: Effect sizes and p-values for DMC analysis (cg03546163 and asthma), eQTM analysis (cg03546163 and FKBP5 expression) and DEG (FKBP5 expression and asthma) analysis pre- and post-adjustment for methylation at the CpG (labeled DEG_{unadj} and DEG_{adj}). **Panel C:** UCSC Genome Browser view of the FKBP5 locus, indicating locations of cg03546163 (pcHiC) and cg23416081 (5kb of TSS) showing interaction between the GeneHancer regulatory elements at these two regions. Publicly available data from tracks displayed includes location of exonic and intronic gene regions from the UCSC gene annotation; regulatory elements, genes and their interactions from GeneHancer, in detailed

and clustered views; chromHMM tracks from Roadmap; transcription factor CHIP-seq from ENCODE; and DNase hypersensitivity density signal from ENCODE for CD20+ B-cells, CD14+ monocytes, fibroblasts and naïve B-cells.

Figure S6: Integrative analysis of methylation and gene expression at *TREML2* and *TMEM71*. **Panel A:** Distribution of gene expression and cpg methylation by asthma status. **Panel B:** Summary of association effect sizes and p-values relating gene expression, methylation and asthma. Results are shown for DMC analysis, eQTM analysis and DEG analysis pre- and post-adjustment for methylation beta at the CpG (labeled DEG, $_{unadj}$ and DEG, $_{adj}$), to determine if there is an association between gene expression of *FKBP5* and asthma independent of methylation. Upper row is *TREML2* and bottom row is *TMEM71*.

(6) Figure 2C, three modules (m6, m5 and M4) expression are used for plotting case control status. However, the three modules do not clearly separate cases from the controls. Do you check other modules? For example, M3 had high FC = 0.16 instead of using M4 (FC = -0.13). I expect clear separation of the cases and controls in Fig 2C and higher OR in Fig2D.

Response: We did not check all modules as there is correlation between modules themselves (evident in Figure 2A: M6 & M4: $R = -0.70$, $p < 0.001$ / M5 & M6: $R = -0.58$, $p < 0.001$ / M5 & M4: $R = 0.29$, $p < 0.001$). However, our selection of the three modules was not arbitrary; we selected the most significant module of each axis of dysregulation based on q-value in Table 1: specifically M6 was selected for *Th2 inflammation*, M5 for *airway remodeling* and M4 for *drug response*. We have now clarified this in the methods section for the report.

With regards to separation we note that much higher odds ratios would be needed for a clear separation of cases and controls. To demonstrate this, we consider a hypothetical example where scores for cases and controls come from overlapping normal distributions. It is subjective what “clear separation” means, but in the left panel below we show an example where the shift in means was chosen to produce 90% sensitivity and 90% specificity if the cut-off for classifying an observation was chosen as the midpoint (dotted line). The resulting odds ratio is $(0.9/0.1) / (0.1/0.9) = 81$. In our manuscript, the largest odds ratio (comparing 3 modules versus 0 modules) is 4.95. This corresponds to a shift in means yielding about 69% sensitivity and specificity, shown in the right panel. This degree of separation appears consistent with the findings reported

in our manuscript. A fold change of 0.16 versus a fold change of 0.13 does not result in a clear separation. In our logistic model that relates the number of modules to the fraction of asthma we also conducted a formal test for departure from additivity in the number of modules allowing for two additional degrees of freedom. The likelihood ratio test yielded a p-value of 0.29. Thus, we picked these three modules for accuracy and ease of interpretation.

Finally, to address **point #33** below by **Reviewer #3**, we have eliminated the prior panel 2C.

Changes: See **Figure 2** for removed 3D panel.

(7) The study (including the title) presented as African diaspora. However, the samples include diaspora and non-diaspora. It is not scientifically accurate to characterize population from different regions as diaspora. I suggest removing diaspora from the title.

Response: *Population geneticists refer to our species as one species with an African origin; therefore, we are all part of the African Diaspora. There have been multiple African Diasporas, the most recent spurred by European colonials, for which enslaved Africans were brought to the Americas over a 400 year period of time. Populations from the Americas today descend from (at least) indigenous, African and European ancestors and can include peoples with origins all over the rest of the world. Therefore, we contend the African diaspora title is accurate as it includes the multiple waves of African ancestry that are carried by people on the Americas continent and the islands of the Caribbean from which our participants descend.*

Changes: none.

(8) The 4th paragraph of the introduction should not be part of the introduction. It is about methods and results.

Response: We have edited this paragraph to provide an overview but not specific details. We appreciate this suggestion by the reviewer.

Changes:

Introduction: *We hypothesize that transcriptomic signatures from the nasal airway epithelium in asthma cases and controls representing the African Diaspora will allow us to validate previously identified gene expression signatures of asthma and, importantly, identify pathways of dysregulation that are novel and relevant to the disparities observed with respect to asthma. We rely on nasal epithelium as a proxy for the airways given its ease of tissue collection on large numbers of individuals and the established correlation between signatures of asthma between nasal epithelium and bronchial tissue. RNA sequencing and DNA methylation data from the same nasal epithelium samples in cases (individuals with current asthma) and controls (individuals never having asthma) representing 7 locations across the African Diaspora revealed dysregulation on three axes – increased Th2 inflammation, decreased capacity for wound healing in airway epithelium, and impaired drug response – that play a role in the development of asthma in individuals of African ancestry.*

(9) The study lacks multiple testing correction. The liberal significance threshold was used. However, it is not clear what cut off and why. Otherwise, the analysis is descriptive and subjective.

Response: We have applied stringent multiple testing as follows: FDR is used for all RNASeq DEG analysis and the DMC analysis uses Bonferroni corrections. The only analysis in which we rely on uncorrected p-values is the eQTM analysis where the goal was to identify a set of CpGs that are associated with gene expression to carry forward to DMC analysis at which point we then applied a Bonferroni correction. This was previously stated in methods sections, but we have also updated captions in Table S7 and S8 to make this clear. Supplementary tables list p-values and corrected p-values (either q-value or Bonferroni) in the headers to clarify multiple testing criteria and show uncorrected and corrected results (e.g. p-values and q-values).

Changes: See **Table S7 legend** and **S8 legend**.

Methods: There were 8,418 eQTM tests performed for gene-CpG pairs and significance for the eQTMs was defined as eQTM $p < 0.05$; these tests were not corrected for multiple testing as the purpose was to determine a set of CpGs to move forward to DMC analysis with asthma.

(10) Lifestyle, environmental exposure should be included as covariates before comparing sites.

Response: The lack of information on environmental risk factors and social determinants of health in asthma is a limitation of the parent CAAPA study design. Despite this limitation, we are uniquely positioned to investigate asthma multi-omics in populations that are historically under-represented in genomics research but bear a disproportionate burden of the disease and disease severity. We added acknowledgement of this limitation to the discussion section.

Changes:

Discussion: Limitations in our study include the inability to differentiate gene expression profiles of atopy from asthma given the high prevalence of atopy in our cases and controls, the restriction of methylation data to only the US-based recruitment sites, and our inability to investigate environmental risk factors and social determinants of health in asthma. Despite these limitations, we are uniquely positioned to investigate asthma multi-omics in populations that are historically under-represented in genomics research but bear a disproportionate burden of the disease and disease severity. Overall, we found $N=389$ differentially expressed genes, and 16 differentially expressed modules that are associated with current asthma adjusting for differences by site and ancestry. Novel in our findings are the strong signatures related to wound healing and drug response at single-gene and network-based levels that may have identified additional endotypes for asthma with potential implications for targeted therapy in the future.

(11) It is not clear if generating genes based on significant DEGs followed by WGCNA is better or directly running WGCNA is appropriate.. Comparison will be useful.

Response: We appreciate this point and would agree that there is no ‘better’ choice but rather the choice should be dependent on the purpose of the analysis. Here, our purpose for the WGCNA was to identify meaningful networks from identified DEGs. Therefore, our analysis was limited to those DEGs with q-value <0.15 ; we believe this intermediate choice between only significant DEGs (q-value <0.05) and all 21K expressed genes is a reasonable selection to search for modules related to asthma as shown by other groups examining RNASeq and DNAm signatures for asthma (PMIDs: 33713771 and 27942592)

Changes:

Methods: The purpose of this WGCNA analysis was to identify networks of genes within DEGs for asthma; a more liberal significance threshold ($q < 0.15$ vs. $q < 0.05$) was used to allow a deeper query across DEGs in a systems biology analysis framework

(12) The replications are not exhaustive and true replication.

Response: We respectfully disagree with the reviewer on the nature of our replication. For the 21,831 genes tested, we performed an exhaustive search for replication in a meta-analysis study of airway epithelium gene expression in asthma. Briefly, Tsai et al performed a meta-analysis of eight independent gene expression studies including both nasal and bronchial epithelium tissue – this is the largest set of

DEGs relying on airway epithelium, including both nasal and bronchial epithelium that we are aware of. We did not limit ourselves to the set of significant ($q < 0.05$) DEGs reported by the authors. Rather, full results from the meta-analysis were obtained directly from these authors and were compared to the CAAPA results by matching genes on Ensembl ID. Additionally, we report replication at the most stringent level: we first look at q-value reported by Tsai et al defining replication as an independent discovery in the meta-analysis, and only secondly at a nominal uncorrected p-value reported by Tsai et al.

Changes: none.

(13) True validation of relevant genes including functional analysis are critically important.

Response: This is beyond the scope of this current work, but we agree that this is a natural follow up to these findings.

Changes: none.

(14) Childhood asthma is different from adulthood asthma. So separate analysis should be followed throughout the manuscript.

Most adult asthma begins in childhood, especially early childhood. In our sample, 75% of all cases, regardless of age at recruitment, have a childhood onset (≤ 12 years), and an additional 8% have an age of onset ≤ 18 years. If one looks at genetic underpinnings for asthma, the identified genetic loci are mostly common between adults and children with respect to the age of diagnoses (i.e. age of onset) of asthma (PMID: 31036433); in fact only one supports genetic underpinnings unique to adult onset. Therefore our approach here is to include all asthma cases to identify shared signatures of asthma; we do not have power to test adult vs. childhood onset separately.

Changes: none.

(15) It is not clear what second phase of CAAPA refers to.

Response: This is our second round of the consortium – in the first phase of CAAPA, the consortium sequenced ~1,000 genomes of African ancestry, helped design genotyping array content to address poor coverage of the then-available GWAS arrays in regards to African ancestry populations, and performed the largest GWAS in African ancestry individuals for asthma. In this phase, the consortium is focused on multi-omics and systems biology approaches to understand the genomic and molecular underpinnings of asthma with a focus in individuals of African ancestry. However, we have removed all reference to phases of the consortium as we have feedback from multiple reviewers that this is not relevant to this report.

Changes:

Removed from Discussion: *In this second phase of the Consortium on Asthma among African ancestry Populations in the Americas (CAAPA), we expanded our efforts to dissect asthma health disparities by defining multi-omics signatures in current asthma cases and never-asthma controls from 7 different sites representing the African Diaspora including Baltimore, Denver, Washington DC and Chicago in the United States, Barbados in the Caribbean, Brazil in South America and Nigeria in West Africa.*

(16) There is no plan to submit the data to the public repository like GEO so that others can replicate the study.

Response: We apologize for the misunderstanding –the GEO submission was in progress at time of first submission as was communicated to the editor. **Data are now available: GSE240567**

REVIEWER #2

This is a well written paper which addresses nasal epithelial gene expression differences between asthma cases and controls in a population of African ancestry subjects. The main conclusions are that pathways involved in wound healing (and in particular a role for FN1) and drug responsiveness are differentially regulated in cases v controls. Additional data trying to address epigenetic regulation are included using eQTM approaches. Whilst many of the findings reinforce the role for a range of genes for which a role in asthma is already well established (eg many genes involved in Th2 responsiveness) there are fewer data available from this ancestry group, and overall therefore the findings add to the current literature. The methodology is sound.

Response: We thank the reviewer for their support; while the role of Th2 pathways is well documented, to our knowledge the strong roles of both wound repair and drug responsiveness are novel findings and these need further follow-up to determine if these play difference roles based on ancestry. We appreciate the recognition that there is minimal data in this underserved group. Specific responses are below in blue:

There are a few points which the authors could consider addressing:

(17) Population characteristics. The populations used are relatively small from each site (the smallest being 62 individuals, see supplementary table 1) and of quite diverse ancestry (see supplementary figure 1a). Whilst this has been factored into the analyses presented, how these differences were dealt with probably deserves some further comment; in particular the epigenetic analyses were only performed in USA subjects.

Response: Thank you for this suggestion, we have added a note on the limitation of the epigenetic data to the US-only sites in the Discussion. We have addressed issues of site differences and ancestry effects through a comprehensive covariate adjustment in all our omics analysis, and have added a note to this as well in the first paragraph of our Discussion.

Changes:

Discussion: Limitations in our study include the inability to differentiate gene expression profiles of atopy from asthma given the high prevalence of atopy in our cases and controls, the restriction of methylation data to only the US-based recruitment sites, and our inability to investigate environmental risk factors and social determinants of health in asthma. Despite these limitations, we are uniquely positioned to investigate asthma multi-omics in populations that are historically under-represented in genomics research but bear a disproportionate burden of the disease and disease severity.

(18) Treatment effects. Many of the subjects in addition to having asthma had other features of allergic disease. How many were taking nasal steroids? The table suggests 61% were using 'as required' or no medication (I presume this refers just to asthma).....did this include ICS or just SABAs? This does matter as the conclusions around FKBP5 being upregulated in asthma could merely be due to medication rather than the underlying disease. Indeed, the effect size for FKBP5 was higher in those on regular higher dose steroids.

Response: Thank you for noting this excellent point. As part of the nasal epithelium sample collection, we gathered information on nasal steroid (NS) use. Participants were asked whether they were on NS; if they were on NS, they were further asked whether they withheld usage in the 5 days prior to nasal epithelium sampling. To address this question, we performed a differential expression analysis comparing cases using NS that did not withhold NS for 5 days to cases on NS that did withhold NS for 5 days to

determine if there was gene expression difference based on the NS usage within the 5 day window of tissue sampling.

Of the N=1,326 DEGs with FDR<0.15 from our main analysis, we did not observe any differential expression by NS usage; minimum q-value in these genes was 0.89. Specifically, for FKBP5, the p-value was 0.18 and corresponding q-value was 0.99. Therefore, our gene expression results are robust to nasal steroid use. The robustness of our DEGs to medication use is also supported by our existing analysis adjusting for ICS usage in the original submitted report (Table S5).

Our findings regarding the robustness of asthma gene expression signatures to ICS or NS usage are similar to evidence for DNAm in airway epithelial cells (PMID27942592) wherein the authors found no difference in methylation signatures in cultured (primary) bronchial epithelial cells exposed to glucocorticoids.

Changes:

Tables: *New Table S6*

Results: *There were 24 WGCNA modules identified from analysis of N=1,326 genes (DEGs with FDR<0.15). Of these, 16 modules ranging in size from 21-88 genes were significantly differentially expressed by asthma status (Table 1, Fig 2); gene-module membership is shown in Table S2. We note that none of these N=1,326 genes were differentially expressed based on nasal steroid (NS) usage 5 days prior to nasal epithelium sampling in cases (minimum q-value = 0.89 comparing 32 cases on NSs but not withholding compared to 36 cases on NS that withheld NS; Table S6).*

* Also see Supplementary Methods

(19) It is reassuring that in the replication analyses for the DGEA 87/353 replicated with for 86 signals this having the same direction of effect in the dataset used (ref 26...note this is bronchial epithelial rather than nasal epithelial but the authors provide support that there is reasonable concordance between the 2 sites in gene expression changes in asthma). The data for FN1 look robust across these data sets. However, if the aim in part was to look for ancestry specific effects, using a replication set which isn't in the same ancestry group may lead to a failure to replicate.

Response: We have now clarified in the methods that our replication study includes studies on gene expression in both nasal and bronchial epithelial tissues, although there are more samples representing bronchial epithelial tissue and that may potentially drive the prior signals. In this current report, it is not our aim to assess ancestry-specific effects (i.e. heterogeneity in DEGs between ancestry groups) as our study design does not include non-African ancestry individuals. Here, we adjust for ancestry and sampling site in testing for DEGs for asthma; we do not examine gene expression differences as a function of African ancestry. With the availability of SNP genotype data genomewide, CAAPA will offer us the future opportunity to perform a detailed exploration of eQTLs and gene expression considering local ancestry as recently shown by Kachuri et al using whole blood gene expression (PMID: 37231098).

Changes:

Methods: *For the 21,831 genes tested in the DEG analysis of all subjects, we searched for replication in a meta-analysis of 8 studies of airway epithelium gene expression in asthma; N=6 using bronchial epithelium, and N=2 using nasal epithelium. Enrichment of CAAPA DEGs in the meta-analysis DEGs was tested using a hypergeometric test. The total number of genes tested was determined as the number of*

unique Ensemble IDs retrieved in the meta-analysis full results that matched to Ensembl IDs included in the 21,831 genes tested in CAAPA.

(20) In the eQTM analyses it looks as if 8418 gene CpG pairs were examined. I assume though that methylation was studied genome wide, so there must be extensive other data available which are not presented in this paper. Given the number of methylation sites on the current chips including genome wide methylation analyses to explore trans effects would present a challenge without a replication population so I think this is acceptable. 918/8418 gene CpG pairs were considered significant at $p=0.05$ although many of these could be false positives given the low stringency p value used: this probably deserves comment. Only 5 DMCs survived multiple testing correction, but the FN1 locus was not one of these. What do the authors infer about these relationships....do they think they may be causal, or due to the effect of the presence of disease, or due to drug treatment (I note that FKBP5 comes up here)?

Response: We appreciate the reviewer's support of our rationale for the inclusion of the DNAm data for follow up on the identified DEGs. They raise a fair point that we have applied a less stringent threshold on the eQTMs, but our approach here was to identify a set of target CpGs to then test as DMCs, and there we do apply a Bonferroni threshold. We address this in our response to Reviewer #1 as well, and we have now added this to the methods section of the report. We agree with the reviewer that there are many other mechanisms of regulation of gene expression beyond DNAm, and therefore it should not be unexpected that we only find these coalescent signatures for a small subset of genes under stringent thresholds for multiple-correction.

As pointed out in our results and discussion - while site-specific effects of gene expression at *FKBP5* with asthma are consistent (with overlapping confidence intervals), a markedly stronger effect was observed in the Brazil site which is also the site with the most severe asthmatics and highest ICS use in asthma cases. And yet, *FKBP5* is not differentially expressed by ICS use in the full CAAPA dataset ($p = 0.615$). As pointed out above, differential expression at *FKBP5* is also not related to NS usage. When limited to those asthma cases not on moderate-high ICS use ($N=283$), the association with asthma remained ($\beta = -0.197$, $p = 1.04 \times 10^{-3}$). We also observed a more significant association between methylation and asthma severity ($\beta = -0.393$, $p = 8.17 \times 10^{-4}$), than between methylation and ICS usage ($\beta = -0.314$, $p = 0.014$). Unfortunately, our study is limited in its ability to evaluate causality here, but our overall interpretation is that our results show that the *FKBP5* findings may be in part explained by the medication use, but it is not possible to rule out the hypothesis that *FKBP5* expression levels led to asthma severity, which then leads to the increased medication use. We try to capture this in the discussion focusing on the Brazil and Denver sites.

Changes: none.

REVIEWER #3

Summary: Szczesny et al obtained RNA-Seq and Methylation data for nasal epithelium samples from 253 asthma cases and 283 never-asthma controls from a cohort called Consortium on Asthma among African-ancestry Populations in the Americas (CAAPA). Analyses included differential gene expression, pathway analysis, gene co-expression network analysis and eQTLs and differential methylation sites for select genes. Some analyses included fewer subjects (eg DNA methylation and RNA-Seq was 298 total). They found various genes implicated in asthma via these analyses, including glucocorticoid responsive genes. A few lines of results are interesting (FN1 as top differentially expressed, upstream regulators, FKBP5 and methylation results integrated with differential expression) but they are not functionally validated or linked in such a way that provides compelling new insights into asthma or its related processes. Most results presented are then supported by prior literature which strengthens likelihood that results are generalizable while also diminishing novelty of paper given that hardly any findings remain truly novel. Loosely collating the results of analyses into three axes seems to be an attempt to make a cohesive story out of a few lines of research that are not fully pursued. The results may nonetheless interest others in the field who may want to corroborate their own findings or pursue experiments based on the hypotheses generated by the work presented.

Response: We thank the reviewer for their feedback. We have made an attempt to address all major criticisms below, and hope that this addressed their primary concerns.

Major

(21) The introduction and discussion bring up “In this second phase of CAAPA,” but it is not clear what the first phase was – the GWAS of a prior introduction paragraph? Or, is there something else that is being referred to? How many phases does CAAPA have?

See point #15 from Reviewer #1 above.

(22) Referring to three processes as axes implies independence among them that likely does not reflect biological processes given that Th2 inflammation, wound repair and drug responses to steroids and other asthma drugs are all interrelated. More clarity around the limitations implicit in taking a cross sectional view that is displayed by the modules should be provided as the processes are not truly independent.

Response: We make no assumption that axes are independent nor do we make conclusions to this effect. In fact, our Figure 2A is used to show that the modules are correlated (M6 & M4: $R = -0.70$, $p < 0.001$ / M5 & M6: $R = -0.58$, $p < 0.001$ / M5 & M4: $R = 0.29$, $p < 0.001$). Our selection of the three modules was not arbitrary; we selected the most significant module of each axis of dysregulation based on q-value in Table 1: specifically M6 was selected for *Th2 inflammation*, M5 for *airway remodeling* and M4 for *drug response*. We have now clarified this in the methods section for the report. To further enhance our transparency in this matter, we have now added the correlation between the three modules selected and make the point that these are not indeed independent.

Changes:

Methods: *We selected the most significant module based on q-value for Th2 inflammation (M6, CEACAM5), wound repair (M5, FN1), and drug response (M4, NCALD).*

Results: *The WGCNA modules reflected the same three axes of dysregulation in asthma cases as indicated from the top 15 DEGs and IPA upstream regulators on 389 DEGs; it should be noted that many*

of the associated modules and therefore the axes of dysregulation are not independent (**Fig 2A**). The cumulative effect of these three axes is illustrated in a joint model examining dichotomized module expression focusing on the most significant module for each axis: Th2 inflammation (CEACAM5,M6), wound repair (FN1,M5), and drug response (NCALD,M4) (**Fig 2C**). These modules are significantly correlated (M6-M4: $R = -0.70$, $p < 0.001$ / M5-M6: $R = -0.58$, $p < 0.001$, and M5-M4: $R = 0.29$, $p < 0.001$).

Discussion: This work reveals dysregulation of three axes – increased Th2 inflammation, decreased capacity for wound healing, and impaired drug response. Each is associated with risk for asthma and there is correlation between the axes themselves, but the impact of dysregulation on multiple axes bears a cumulative risk with an OR of 4.95 (95% CI = 3.09-7.93).

(23) Multi-module analysis is a bit arbitrary as presented. Were individual modules for the traits in question determined on the basis of single genes. For the WGCNA analysis, the modules are based on groups genes for which the authors do not say much. Representing each module as a single gene as in Table 1 (or no genes as is the case for some) is not helpful for. An ontological enrichment analysis of the genes in each module, at least for the top modules, may help with interpretation. Similarly, listing some of the many genes within them can help with interpretation, at least for some top genes that are differentially expressed.

Response: We apologize that our selection criteria for the three modules were not transparent. We selected the most significant module based on q-value of its association with asthma within each category – one for Th2, one for drug response and one for airway remodeling. The modules were selected based on their overall association with asthma, and not any single gene, although WGCNA was run here on a subset of genes (those DEGs identified at FDR <0.15). The membership of genes, i.e. the full listing of all genes belonging to each module are specified in Table S2. Furthermore, we have not limited our naming of each module to a single final gene; we have a full listing of hub genes (either through WGCNA or STRING) and most significant DEG within the module in Table 1.

Changes: See modified **Table 1**.

Methods: We selected the most significant module based on q-value for Th2 inflammation (M6, CEACAM5), wound repair (M5, FN1), and drug response (M4, NCALD).

(24) Gene expression signatures by their nature are highly variable and dependent on many environmental and internal factors. Thus, while it is impressive that the authors collected samples from people from 7 different geographic locations, having only ~500 people represented is a limitation to establish generalizability. Based on results from prior transcriptomic studies of asthma, there is no clear asthma “signature” and cluster analysis has not yielded highly reproducible groups aside from the Th2 high and low ones. There are many differences across sites in terms of severity, drug use, age, etc, which also confound relationships observed. This needs to be described as a major limitation of the work.

Response: We respectfully disagree with the concern around generalizability; we would counter that if our study had included asthma cases and controls from a single site we would have a greater challenge with generalizability. Our analysis pipeline carefully considered the issue of potential confounding by site with non asthma-related gene expression signatures from the point of plate balancing (including case/control/site/age/sex randomization) to the inclusion of technical assay covariates and sampling site covariates. Our identified genes are robust to site-differences: for example (1) as noted in our volcano plot

in Fig 1A most of the identified DEGs are significant in multiple site strata; and (2) the top 15 DEGS have generally consistent effect sizes as noted in Figure S2 with overlapping confidence intervals. We do not attempt to cluster individuals in our analysis, and agree that had we done that we would have had several major limitations including power. Our WGCNA approach is used to cluster co-expression genes into networks and we then look at the relationship between average expression of networks and asthma. That said, we have added a limitations section in the paper to speak to several points raised by the reviewers, and we have addressed the issue of confounding in our methods, results and discussion.

Changes:

Discussion: Limitations in our study include the inability to differentiate gene expression profiles of atopy from asthma given the high prevalence of atopy in our cases and controls, the restriction of methylation data to only the US-based recruitment sites, and our inability to investigate environmental risk factors and social determinants of health in asthma. Despite these limitations, we are uniquely positioned to investigate asthma multi-omics in populations that are historically under-represented in genomics research but bear a disproportionate burden of the disease and disease severity. Overall, we found N=389 differentially expressed genes, and 16 differentially expressed modules that are associated with current asthma adjusting for differences by site and ancestry. Novel in our findings are the strong signatures related to wound healing and drug response at single-gene and network-based levels that may have identified additional endotypes for asthma with potential implications for targeted therapy in the future.

(25) Table 1 should include participant characteristics given that this is key information for the reader to interpret study design and findings, and not be relegated to the supplement.

Response: Thank you for this suggestion. We wholeheartedly agree with this suggestion but have currently placed this as a supplement given limitations by the journal on tables/figures allowed. We will discuss this with the editorial team and make every effort to address this comment if accepted.

Changes: to be determined with editorial office.

(26) Having DNA methylation only from four sites, all in the US, is another limitation that introduces bias in the results from this modality.

Response: We agree that this is a limitation, but performing the additional arrays was beyond the cost scope of this project. We have noted this as a limitation.

Changes:

Discussion: Limitations in our study include the inability to differentiate gene expression profiles of atopy from asthma given the high prevalence of atopy in our cases and controls, the restriction of methylation data to only the US-based recruitment sites, and our inability to investigate environmental risk factors and social determinants of health in asthma.

(27) For testing of 8418 eQTM, the significance threshold ($p < 0.05$) was not corrected for multiple comparisons made. This suggests those results are less reliable than any others.

See point #9 from Reviewer #1 above.

(28) Typo line 260: “The effect of methylation was evaluated looking at the change in effect size and significance of the DEG between the DEG,unadj and DEG,unadj models.”

Response: Noted, and fixed.

(29) Replication occurring in airway epithelium is a limitation. Although nasal and airway are similar, they are not the same. This is not mentioned in discussion section.

Response: For the 21,831 genes tested in the DEG analysis of all subjects, we performed replication in a meta-analysis study of airway epithelium gene expression in asthma. This meta-analysis by Tsai et al included eight independent gene expression studies *in both nasal and bronchial epithelium tissue* – this is the largest set of DEGs relying on airway epithelium, including both nasal and bronchial epithelium that we are aware of. This study is described in greater detail in the methods sections for Replication Analysis in the main body and Supplementary methods of the report.

Changes:

Methods: For the 21,831 genes tested in the DEG analysis of all subjects, we searched for replication in a meta-analysis of 8 studies of airway epithelium gene expression in asthma; N=6 using bronchial epithelium, and N=2 using nasal epithelium.

(30) Searching for upstream regulators rather than pathways represented by the 389 DEGs is interesting, but how would no genes that are regulators of the DEG ones not be differentially expressed themselves?

Response: We used IPA on the N=389 DEGs identified at FDR <0.05 to identify upstream regulators, but we used WGCNA on the N=1,326 DEGs identified at FDR <0.15 for networks based on correlations between genes. There are 16 DEGs (FDR<0.05) with an IPA p-value of overlap from the upstream regulator analysis including FN1; we regret that we were not clear in our presentation of results our Figure S3 and Table S4 only named the top 10, although Figure S4 shows all significant ones with red dots.

The idea that upstream regulators are themselves not DEGs or not top ranked DEGs is not unexpected: there are several existing examples of this in asthma (see comment on IL4 and IL13 below). Furthermore, IPA acknowledges this very point for their upstream regulator analysis: "*The analysis does not take into account the gene expression observed for the predicted upstream regulator itself, because the gene expression for the upstream regulator may not differ between experimental and control samples. For example, the regulator may be expressed in adjacent cells or may be activated by other means rather than by increased expression. The analysis predicts the activity of the upstream regulator's encoded protein (or mature microRNA).*" Please see: <https://qiagen.my.salesforce-sites.com/KnowledgeBase/articles/Knowledge/Upstream-Regulator-Analysis>

Comment on IL4 and IL13 – genes encoding ‘key pathophysiologic cytokines and druggable molecular targets’ (PMID: 35350765) in asthma are hallmark cytokines with upstream effects in multiple pathways in asthma. Neither is sufficiently expressed in our data to be included in our analysis and they were

filtered out as the DESeq2 mean normalized count was below 20. The meta-analysis by Tsai et al does not find IL4 to be a DEG (q-value= 0.93) for asthma. The same meta-analysis finds IL13 to be a DEG (q-value= 0.01), but it is only the 868th ranked DEG. Despite the lack of (IL4), or modest (IL13) DEG evidence, these are highly targeted cytokines in asthma management precisely because of their downstream effects: IgG4 monoclonal antibody dupilumab selectively blocks IL-4R α and works as antagonist of both IL-4 and IL-13. Therefore looking at upstream regulators is an important aspect to understanding the relationship between our DEGs and asthma, irrespective of whether or not they are themselves identified as DEGs.

Changes: We have now added the IPA results for all upstream regulators with $p < 0.05$ to **Table S4**, and it can be seen that 16 of our DEGS are also identified as upstream regulators, including FN1. Additionally the legend to **Table S4** now has been edited to say this information is limited to the top 10.

(31) Along those lines, glucocorticoid use and responsiveness is likely different among subjects and impacts the DEGs as suggested by IPA analysis. How do authors explain that no genes regulated by the drugs are differentially expressed yet the upstream regulators are significant? FKBP5 is a canonical marker of glucocorticoid response in vitro, with levels that rise shortly after exposure and are maintained for at least 24 hours. Suggesting that levels of such a gene that vary over short time periods are stable biomarkers is a stretch despite some evidence from a publication that it could be a drug response biomarker. Without knowing the time frame when a person took medications, it is difficult to conclude much.

Response: To clarify, all the targets for dexamethasone and fluticasone are DEGs – see methods lines 221-222 where we state that IPA was run on the N=389 DEGs. However, they are only not related to ICS use directly. Our findings regarding the robustness of asthma gene expression signatures to ICS usage is similar to evidence for DNAm in airway epithelial cells (*PMID: 27942592*) wherein the authors found no difference in methylation signatures with treatment.

Additionally see detailed response and new analysis on nasal steroid use in **point #18** from **Reviewer #2** above.

(32) Sentence starting on line 326 mentions 879 differentially expressed genes, in contradiction with number used for IPA

Response: WGCNA was run on N=1,326 genes that had an FDR<0.15 in the DEG analysis. We identified 24 modules from these genes. Of these, median gene expression for 16 modules was significantly different by asthma status, and these 16 modules included N=879 genes. We have edited this section to avoid confusion.

Changes:

Results: There were 24 WGCNA modules identified from analysis of N=1,326 genes (DEGs with FDR<0.15). Of these, 16 modules ranging in size from 21-88 genes were significantly differentially expressed by asthma status (Table 1, Fig 2); gene-module membership is shown in Table S2.

(33)) Figure 2 panel C does not convey any strong results given the large scatter of points from

cases/controls across the 3D space. Choosing a 3D plot to be represented in 2D is usually not helpful as this plot demonstrates. Panel D perhaps makes a better case although there is some tautology in showing that more the groups defined by the differentially expressed genes used to define the groups confer differences in asthma risk.

Response/Changes: We have edited Fig 2 to remove panel C.

(34) Figure 3 lacks clarity in labels for Panel A (especially for box plots), message conveyed for Panel B, and sources of data for Panel C. For Panel C, please clarify that nothing displayed corresponds to authors findings. Legend mentions general data represented, but what are the many tracks shown and why were they selected?

Response/Changes: Thank you for pointing this out, we have edited Figure 3 legends to be clearer.

Changes:

Figure 3: Epigenetic mechanism relating gene expression to asthma for FKBP5. Panel A: Scatter plot of methylation (beta) values at cg03546163 vs gene expression (log2 CPM) values for FKBP5 and box plots showing median and interquartile range by asthma case and control status for N=298 individuals. **Panel B:** Effect sizes and p-values for DMC analysis (cg03546163 and asthma), eQTM analysis (cg03546163 and FKBP5 expression) and DEG (FKBP5 expression and asthma) analysis pre- and post-adjustment for methylation at the CpG (labeled DEG_{unadj} and DEG_{adj}). **Panel C:** UCSC Genome Browser view of the FKBP5 locus, indicating locations of cg03546163 (pcHiC) and cg23416081 (5kb of TSS) showing interaction between the GeneHancer regulatory elements at these two regions. Publicly available data from tracks displayed includes location of exonic and intronic gene regions from the UCSC gene annotation; regulatory elements, genes and their interactions from GeneHancer, in detailed and clustered views; chromHMM tracks from Roadmap; transcription factor CHIP-seq from ENCODE; and DNase hypersensitivity density signal from ENCODE for CD20+ B-cells, CD14+ monocytes, fibroblasts and naïve B-cells.

(35) Claims about identification of endotypes and implications for targeted therapy should be removed given that authors did not explore or confirm clusters, nor do they have an appropriate sample size to do so.

Response: We have removed claims on therapeutic targets from our discussion. However we do feel that the potential for additional endotypes for asthma are fair speculative points based on our novel findings related to wound healing and drug response.

Changes: See final paragraph of Discussion for deleted section.

Deleted: *Moreover, randomized clinical trials continue to bias on populations that do not fully represent the U.S. patient population, leading to therapeutics that are potentially less effective in underrepresented populations, despite those populations suffering greater morbidity and mortality of disease. The findings from our study are significant because they suggest pharmacogenomic targets that may represent distinct asthma endotypes which could inform optimal triaging for specific treatment regimens. Importantly, the identification of these targets was generated from an ancestry group notoriously disenfranchised from clinical trials and for whom standard of care has arguably reflected the status quo.*

(36) Discussion does not include clear listing of limitations, of which there are many important ones. Instead, the final paragraph is overly optimistic about results being helpful for endotyping, finding new drugs, asthma biologics, etc.

Response: We have added comments of limitations of this study in our Discussion section.

Changes:

Discussion: Limitations in our study include the inability to differentiate gene expression profiles of atopy from asthma given the high prevalence of atopy in our cases and controls, the restriction of methylation data to only the US-based recruitment sites, and our inability to investigate environmental risk factors and social determinants of health in asthma. Despite these limitations, we are uniquely positioned to investigate asthma multi-omics in populations that are historically under-represented in genomics research but bear a disproportionate burden of the disease and disease severity.

Minor

(37) Typo line 149: therapy

Response: Noted and fixed.

REVIEWER COMMENTS

Reviewer #1 (Remarks to the Author):

The authors have addressed most of my comments. However, there are some remaining.

1. The finding in the abstract started with "CAAPA represents diversity across the African Diaspora with a wide range of continental African ancestry (9%-100%)". However, neither the introduction method section mentions the ancestry finding that will be reported as the first finding.

2. It was not clear what types of inhaled corticosteroids used from each recruitment center including various places in the US, Brazil, Nigeria Barbados). Do the USA, Africa and Brazil have same guideline for IC.

3. The omics interaction should be based on 4 sites that have both RNAseq and DNAm (not 7 sites of RNA seq and 4 sites of DNAm). This is because the 389 DEGs were generated based on 7 sites not 4.

4. The argument of "these tests were not corrected for multiple testing as the purpose was to determine a set of CpGs to move forward to DMC analysis with asthma" was not clear. I'm not sure lists of false-positive CpGs have scientific merit.

5. The authors do not mention how many pathways were tested in IPA. It is not clear if the IPA results corrected for multiple testing.

6. I don't consider this as a true replication experiment. Rather this is a simple look up from the previous publication perhaps with different inclusion and exclusion criteria.

7. For the sake of power, I strongly disagree with combining childhood asthma and adulthood asthma. Gene expression and DNA methylation greatly differ between children and adults due to lived experience and hence the amount of exposure to the environment. As we know, DNA methylation serves as surrogate for environmental exposures.

8. Figure S1 is not the focus of this publication. If was based on previous work, it could be simply referenced. Otherwise, detailed methods should be provided.

9. The authors were responsive to depositing the RNA seq data to GEO but it was not clear why DNAm data is equally deposited. In order the scientific community to replicate or validate, both RNAseq and DNAm used in this study need to be deposited.

Reviewer #2 (Remarks to the Author):

The authors have revised this manuscript and in the response to reviewers document have addressed all the issues initially raised in my first review: in particular they have undertaken additional analyses looking at the effect (or lack thereof) of nasal steroids on gene expression etc.

I could not find a marked up version of the manuscript which made reviewing the changes more difficult (apologies if I missed it amongst the various files) but have checked the changes made in the clean version and the changes indicated in the response to reviewer comments document appear to have been made in the clean manuscript.

Some of the limitations identified in this study remain, but they have been discussed adequately in the revised version

Reviewer #3 (Remarks to the Author):

The authors have adequately addressed my concerns.

The authors have addressed most of my comments. However, there are some remaining.

1. The finding in the abstract started with “CAAPA represents diversity across the African Diaspora with a wide range of continental African ancestry (9%-100%)”. However, neither the introduction method section mentions the ancestry finding that will be reported as the first finding.

8. Figure S1 is not the focus of this publication. If was based on previous work, it could be simply referenced. Otherwise, detailed methods should be provided.

Response: This is a necessary descriptive part of the paper as it describes ancestry representation in our representative sample from each geographic site. The results are not based on prior work, but the sample at hand. We thank the reviewer for pointing out that we left out the methods corresponding to this section which may have resulted in the confusion. This has now been added.

Changes to paper:

Main Manuscript: Lines added 188-196

“Quantification of principal components and global sample ancestry:

To estimate global ancestry proportions, we first implemented cross-validation using ADMIXTURE 31 to determine the number of reference populations (K) with K=1–5, to infer the optimal number of ancestral reference groups needed. K=3 accounted for the lowest cross-validation error. CAAPA samples were merged with 3 reference populations as detailed in Supplementary Methods. With a set of 219,832 autosomal SNPs obtained after the merge, and using K=3, we performed global ancestry estimation using ADMIXTURE and plotted the admixture estimates using the PONG visualization tool³² as shown in Fig S1. Details on principal components analysis used in further analyses are provided in Supplementary Methods.”

Online Supplementary Methods: Lines added 141-160:

“Quantification of principal components and global sample ancestry:

We used KING 10 to estimate relatedness and build a kinship matrix including all subjects from all the sites. SNPs with >5% missingness, <1% minor allele frequency and hardy Weinberg $p < 1 \times 10^{-6}$ were discarded. SNPs underwent linkage disequilibrium-pruning (removal of SNPs with an R^2 -value >0.1 within every 50 SNP window) and PCA analyses were performed using a CAAPA-only dataset comprised of 673 samples and 512,925 SNPs. PCA was performed allowing for observed kinship using PC-AiR as implemented in the GENESIS R package¹¹. Additionally, PCA was also performed including reference populations from the 1000 Genomes Project (85 Utah residents with Northern and Western European ancestry CEU (EUR), 88 Yoruba samples from Ibadan, Nigeria YRI (AFR) and 43 Native Americans selected from Mao et al. 12 (AMR)) on 219,832 autosomal SNPs obtained after the merge with reference data. The elbow in the scree plot was used to identify the top two PCs (PC1 and PC2) as covariates for ancestry adjustment, Fig S1.

To estimate global ancestry proportions, we first implemented cross-validation using ADMIXTURE 13 to determine the number of reference populations (K) with K=1–5, to infer the optimal number of ancestral reference groups needed. K=3 accounted for the lowest cross-validation error. CAAPA samples were merged with 3 reference populations and with the set of 219,832 autosomal SNPs obtained after the merge, and using K=3, we performed global ancestry estimation using ADMIXTURE and plotted the admixture estimates using the PONG visualization tool¹⁴ as shown in Fig S1.”

2. It was not clear what types of inhaled corticosteroids used from each recruitment center including various places in the US, Brazil, Nigeria Barbados). Do the USA, Africa and Brazil have same guideline for IC.

Response: The guidelines for asthma management and inhaled corticosteroid (ICS) usage have been well established through global guidelines that apply to all sites participating in the CAAPA study (Global

Initiative for Asthma: Global Strategy for Asthma Management and Prevention, 1993-2023, updated annually)¹, and the types of ICS available to all participants in the different recruitment centers/countries are confirmed to be generically similar based on our questionnaire data.

¹ 2023 GINA Report, Global Strategy for Asthma Management and Prevention.
<https://ginasthma.org/2023-gina-main-report/>

Changes to paper: none

3. The omics interaction should be based on 4 sites that have both RNAseq and DNAm (not 7 sites of RNA seq and 4 sites of DNAm). This is because the 389 DEGs were generated based on 7 sites not 4.

Response: To clarify, we did not perform any interaction analysis in this paper, only performed analysis to determine change in association between gene expression and asthma outcome after adjusting for methylation. While the initial set of 389 DEGs was determined using the RNA-seq data from the full set of 7 sites, all further integrative analyses of DNAm and RNA-seq data were indeed only based on data from the 4 sites with both omics data types. This includes the eQTM analyses, differential methylation analyses, and conditional analyses with DNAm included in DEG models. Moreover, for the conditional analyses, we in fact *first* repeated the DEG model on just the subset of RNA-seq data from the 4 sites with both omics data types, and only then assessed changes in the strength of differential expression after conditioning on methylation levels on this data subset. To note all sample sizes are included in each paragraph of the methods section “**DNA Methylation and multi-omics analysis**” reflecting this approach.

Changes to paper: none

4. The argument of “these tests were not corrected for multiple testing as the purpose was to determine a set of CpGs to move forward to DMC analysis with asthma” was not clear. I'm not sure lists of false-positive CpGs have scientific merit.

Response: We appreciate the reviewer’s comment, publishing a list of false positives would indeed not have much scientific merit. However, the set of CpGs we carried forward was simply a means to an end, to generate a subset of candidates for the DMC analysis rather than testing all CpGs. The same concept is applied for example in every RNA-seq analysis. Genes or transcripts are filtered based on the number of reads mapped (e.g., counts per million, CPM) with very low CPM genes being discarded. This generates a subset of genes to be tested for differential analysis, with most of those genes not expected to be differentially expressed (“false positives”). In a low-powered GWAS it is prudent to only test SNPs that have an appreciable minor allele frequency (MAF), as low MAF SNPs will not have enough power to yield genome-wide significant p-values. The key in all of these instances is that the filter applied does not affect the type I error of the subsequent test. The filter simply generates a shorter list of units to be tested, with the idea that the biologically relevant units will be easier to detect among such a shortened list.

Changes to paper: none

5. The authors do not mention how many pathways were tested in IPA. It is not clear if the IPA results corrected for multiple testing.

Response: Ingenuity pathway analysis was used to identify relevant upstream regulators given differential expression values of DEGs identified at $q < 0.05$ in the full dataset. The IPA tool constructs many possible upstream regulator networks and scores regulators by statistical significance based on a causal network derived from the Ingenuity Knowledge Base. This network is based on literature-curated biological findings about compounds and their interactions. This network contains ~40,000 nodes representing mammalian genes and their products, chemical compounds, microRNAs, and biological functions. These nodes are connected by edges representing the experimentally observed relationships relating to expression, transcription, activation, molecular modification and transport, and binding events. The ‘enrichment score’ (Fisher’s exact test p-value) measures overlap of observed and predicted regulated gene sets. The z-score assesses the match of observed and predicted up/down regulation patterns. This analysis measures activity of regulators via measurement of genes known to be differentially expressed by it in a defined direction. This differs from pathway overlap analysis where there is no guarantee that pathway members are differentially expressed upon pathway activation/inhibition. The scores are not corrected for multiple testing after they are generated.

Changes to paper: none

6. I don’t consider this as a true replication experiment. Rather this is a simple look up from the previous publication perhaps with different inclusion and exclusion criteria.

Response: We have been entirely transparent about the replication look up. We have used the largest resource of available DEGs from airway and nasal epithelium for this resource. Using external studies as replication sources for expensive GWAS NGS technology data like WGS and RNASeq is a field standard.

Changes to paper: none

7. For the sake of power, I strongly disagree with combining childhood asthma and adulthood asthma. Gene expression and DNA methylation greatly differ between children and adults due to lived experience and hence the amount of exposure to the environment. As we know, DNA methylation serves as surrogate for environmental exposures.

When using linear models for hypothesis testing the power depends on the non-centrality parameter of the t distribution (or F distribution, respectively, if normality of the parameter estimate is assumed). Larger non-centrality parameters yield larger power. For a given design matrix, this non-centrality parameter depends on the effect size and the sample size. If fold changes in differentially expressed genes were in opposite directions between the childhood and adult asthma groups, we certainly agree with the reviewer that combining those groups would be a bad idea. If fold changes in differentially expressed genes are in the same direction, power can be improved by combining the groups (even if the fold changes are not identical) since a larger sample size means a larger non-centrality parameter. To shine more light on this issue, we conducted two follow-up analyses. We compared the z-statistics (estimated log₂ fold changes divided by their estimated standard errors) from the joint analysis with the z-statistics from a meta-analysis of the stratified analyses using inverse variance weighting, and found that these z-statistics were indeed very similar ($r=0.90$). We also conducted a formal analysis of effect size heterogeneity examining the differences in log₂ fold changes. The resulting p-value distribution across all 21,454 genes that had data for both childhood and adult asthma groups did not indicate significant differences in fold changes (the smallest Bonferroni-corrected p-value was 0.085). Nonetheless, we do have the stratified analysis presented in their entirety in Table S2 to allow any reader to have access to the stratified results.

Changes to paper: none

9. The authors were responsive to depositing the RNA seq data to GEO but it was not clear why DNAm data is equally deposited. In order for the scientific community to replicate or validate, both RNAseq and DNAm used in this study need to be deposited.

Response: The public sharing of data is a central principle of CAAPA. The methylation data has been posted to GEO: GSE250513

Changes to paper: see Data Availability Statement at end of submission.

REVIEWERS' COMMENTS

Reviewer #1 (Remarks to the Author):

The authors address my concern and suggestions.